# VARIABLE COUPLING-ENHANCED LARGE NEIGHBORHOOD SEARCH FOR SOLVING INTEGER LINEAR PROGRAMS

## ABSTRACT

Large Neighborhood Search (LNS) is a heuristic for integer programming that iteratively destroys part of an incumbent solution and repairs it using an ILP solver to efficiently explore large solution spaces. Recent advances in neural LNS have shown strong performance on Integer Linear Programs (ILPs), where graph neural networks (GNNs) learn neighborhood-selection policies under an independence assumption. However, through an example, we identify that the independence assumption ignores variable coupling and assigns equal probability to neighborhoods with vastly different optimization potential. To overcome this limitation, we propose a coupling-enhanced neural LNS (CE-LNS). CE-LNS augments GNN-based neighborhood prediction with graph decomposition to explicitly capture variable coupling, enabling coupling-aware calibration of neighborhood selection. Theoretically, CE-LNS can (i) predict whether constraints are effective or redundant and (ii) refine neighborhood predictions to approximate optimal neighborhoods. Empirically, CE-LNS achieves stronger performance than existing neural LNS frameworks across diverse ILP benchmarks, demonstrating its effectiveness in escaping local optima.

## 1 INTRODUCTION

Combinatorial optimization has a wide spectrum of real-world applications in logistics (Tordecilla et al., 2023; Vadseth et al., 2021; Tordecilla et al., 2023), scheduling (Adams et al., 1988; Dell'Amico & Trubian, 1993; Lenstra et al., 1990), resource allocation (Gavish & Pirkul, 1991; Regaieg et al., 2021; Mystakidis et al., 2024), and network design (Magnanti & Wong, 1984; Gendron et al., 1999; Grötschel et al., 1995), where many problems admit natural formulations as Integer Linear Programs (ILPs). Exact branch-and-bound (B&B) solvers such as SCIP, CPLEX, and Gurobi have seen decades of engineering, yet their exhaustive search can make closing the primal–dual gap impractical on large instances. Large Neighborhood Search (LNS), an effective paradigm for large ILPs, starts from a feasible incumbent and alternates (i) a destroy step that selects a subset of variables to free and (ii) a repair step that re-optimizes only those variables via an off-the-shelf ILP solver.

Designing the destroy policy is pivotal to LNS. Classical heuristics range from fast but weak random selection to informative yet costly Local Branching. Recent learning-based approaches—CL-LNS (Huang et al., 2023), RL-LNS (Wu et al., 2021), and IL-LNS (Sonnerat et al., 2021)—learn destroy policies (via LB imitation, per-variable RL factorization, or contrastive curation) and deliver strong anytime performance across ILP benchmarks. Within the LNS framework for integer programming, whether under reinforcement learning or contrastive learning, the decision space grows exponentially with the problem size. Therefore, most neural LNS frameworks (Huang et al., 2023; Wu et al., 2021; Sonnerat et al., 2021) factorize neighborhood selection into binary choices for each variable—whether the variable is selected for destruction—rather than treating the entire variable subset as a single action. In Section 4, we provide an example to substantiate our theory: two neighborhoods have identical products of per-variable selection probabilities; consequently, any framework based on the independence assumption would assign equal probability to selecting these two neighborhoods. However, one neighborhood admits only a single feasible solution—the current solution—whereas the other contains a strictly better feasible solution.

Neural LNS frameworks represent ILPs as bipartite graphs and employ GNN-based representation learning to output neighborhood selections. However, the expressive power of GNNs is bounded by the 1-Weisfeiler–Lehman (1-WL) test (Xu et al., 2019), which implies that GNNs cannot distinguish between neighborhoods that are equivalent under 1-WL. Since LNS requires rapid decision-making, even though 2-FWL has been shown to approximate strong branching scores (Chen et al., 2024c), its higher-order complexity $O(n^3)$ and substantially larger parameter/data requirements make it unsuitable for LNS. In this paper, we propose CE-LNS. First, a GNN predicts an initial neighborhood. Then, using graph decomposition, CE-LNS refines bag-level representations to model variable coupling. Finally, CE-LNS calibrates the initial neighborhood prediction using the refined outputs. We prove that CE-LNS (1) predicts whether a constraint is active or redundant and (2) effectively calibrates the first-stage prediction to approximate the optimal neighborhood. Experiments show clear improvements over existing neural LNS baselines on both synthetic benchmarks (MVC, MIS, CA, SC) and real-world datasets (WA, IP). Ablations confirm that both the alignment and calibration components are necessary.

## 2 BACKGROUND

**Integer Linear Program.** An integer linear program (ILP) instance $\mathcal{I} = (\mathbf{A}, \mathbf{b}, \mathbf{c})$ is defined as

$$\min_{\mathbf{x} \in \{0,1\}^n} \mathbf{c}^\top \mathbf{x} \quad \text{s.t.} \quad \mathbf{A}\mathbf{x} \leq \mathbf{b}, \tag{1}$$

where $\mathbf{x} = (x_1, \ldots, x_n)^\top$ denotes the $n$ binary decision variables, $\mathbf{c} \in \mathbb{R}^n$ is the vector of objective coefficients, and $\mathbf{A} \in \mathbb{R}^{m \times n}$ and $\mathbf{b} \in \mathbb{R}^m$ specify the $m$ linear constraints.

**Bipartite Graph Representation.** As shown in Figure 3, representing an ILP as a bipartite graph $\mathcal{G}_{n,m} = (V \cup U, E)$ is a mainstream approach for ILP representation learning. The variable nodes $V = \{v_j\}_{j=1}^n$ correspond to decision variables $\{x_j\}_{j=1}^n$, and the constraint nodes $U = \{u_i\}_{i=1}^m$ correspond to constraints. An edge $(v_j, u_i) \in E$ exists if and only if $A_{i,j} \neq 0$, and it carries $A_{i,j}$ as an edge feature.

**Large Neighborhood Search (LNS) for ILP.** LNS is a heuristic that starts from an initial solution and iteratively destroys and re-optimizes a part of the solution until a time limit or a stopping condition is met. Let $\mathcal{I}$ be the input ILP and let $\mathbf{x}^0$ be the initial solution (e.g., obtained by running B&B briefly). At iteration $t \geq 0$, given the *incumbent* $\mathbf{x}^t$ (the best solution found so far), a *destroy heuristic* selects a subset of $k^t$ variables

$$\mathcal{X}^t = \{x_{j_1}, \ldots, x_{j_{k^t}}\}.$$

Re-optimization is performed by solving a sub-ILP in which the variables in $\mathcal{X}^t$ are free while all $x_j \notin \mathcal{X}^t$ are fixed to their values in $\mathbf{x}^t$. The solution to the sub-ILP becomes the new incumbent $\mathbf{x}^{t+1}$, and LNS proceeds to iteration $t + 1$. Compared to B&B, LNS often improves $\mathbf{c}^\top \mathbf{x}$ more effectively on difficult instances Song et al. (2020); Sonnerat et al. (2021); Wu et al. (2021); compared to other local-search schemes, it explores larger neighborhoods per step, helping avoid poor local minima while balancing exploration and tractability.

Following Huang et al. (2023), if iteration $t$ finds an improved solution, we set the *adaptive neighborhood size* $k^{t+1} = \min\{\gamma \cdot k^t, \ \beta \cdot n\}$, where $\gamma > 1$ and $\beta \in (0, 1)$ are constants.

**Local Branching (LB) Heuristic.** Given the incumbent $\mathbf{x}^t$, LB seeks a subset of variables to destroy that yields an optimal $\mathbf{x}^{t+1}$ differing from $\mathbf{x}^t$ on at most $k^t$ variables. Let $h \in \{0, 1\}^n$ be the *neighborhood indicator*, where $h_j = 1$ means the $j$-th variable is destroyed (i.e., $x_j^{t+1}$ may differ from $x_j^t$), and $h_j = 0$ otherwise.

**Neighborhood Search Space.** Given an ILP $\mathcal{I}$, an incumbent $\mathbf{x}^t$, and an indicator $h$, define the neighborhood search space as

$$\mathcal{M}(\mathcal{I}, \mathbf{x}^t, h) = \left\{ \mathbf{x} \in \{0, 1\}^n : \ \mathbf{A}\mathbf{x} \leq \mathbf{b}, \ x_j = x_j^t \text{ whenever } h_j = 0 \right\}. \tag{2}$$

If there exists $\mathbf{x}' \in \mathcal{M}(\mathcal{I}, \mathbf{x}^t, h)$ with $\mathbf{c}^\top \mathbf{x}' < \mathbf{c}^\top \mathbf{x}^t$, we say that $\mathcal{M}(\mathcal{I}, \mathbf{x}^t, h)$ is *effective*.

**Effective / Redundant Constraint.** Let the $i$-th constraint be $\sum_{j=1}^n A_{i,j} x_j \leq b_i$. For a given neighborhood $\mathcal{M}(\mathcal{I}, \mathbf{x}^t, h)$:

- The $i$-th constraint is called *effective* for $\mathcal{M}(\mathcal{I}, \mathbf{x}^t, h)$ if there exists $\mathbf{x}' \in \mathcal{M}(\mathcal{I}, \mathbf{x}^t, h)$ such that $\sum_j A_{i,j} x'_j \leq b_i$ and $\mathbf{c}^\top \mathbf{x}' < \mathbf{c}^\top \mathbf{x}^t$. The effectiveness indicator is $\mathcal{R}(\mathcal{I}, \mathbf{x}^t, h) \in \{0, 1\}^m$.

- Let $\mathcal{I}'$ be the ILP obtained by deleting the $i$-th constraint. If there exists $\mathbf{x}' \in \mathcal{M}(\mathcal{I}', \mathbf{x}^t, h)$ with $\mathbf{x}' \notin \mathcal{M}(\mathcal{I}, \mathbf{x}^t, h)$ and $\mathbf{c}^\top \mathbf{x}' < \mathbf{c}^\top \mathbf{x}$ for all $\mathbf{x} \in \mathcal{M}(\mathcal{I}, \mathbf{x}^t, h)$, then the $i$-th constraint is *redundant* for $\mathcal{M}(\mathcal{I}, \mathbf{x}^t, h)$. The redundancy indicator is $\mathcal{T}(\mathcal{I}, \mathbf{x}^t, h) \in \{0, 1\}^m$.

## 3 INDEPENDENCE ASSUMPTION OF THE PROBABILITY DISTRIBUTION

Inspired by Han et al. (2023), which constructs a probability distribution over solution predictions via an energy function, our goal is to construct a distribution over neighborhood indicators that assigns higher conditional probability to neighborhoods whose induced search space $\mathcal{M}(\mathcal{I}, \mathbf{x}^t, h)$ contains solutions closer to optimality.

Given an ILP instance $\mathcal{I}$ and the current incumbent $\mathbf{x}^t = (x_1^t, \dots, x_n^t)$, the destroy heuristic selects $k^t$ variables to free. Let $h \in \{0, 1\}^n$ be the *neighborhood indicator* with $h_j = 1$ iff variable $x_j$ is destroyed (free) and $h_j = 0$ otherwise; we also write $\|h\|_1 = k^t$. We define the (unnormalized) energy of a neighborhood by the best achievable improvement within its search space:

$$\Delta^\star(\mathcal{I}, \mathbf{x}^t, h) = \max_{\mathbf{x}' \in \mathcal{M}(\mathcal{I}, \mathbf{x}^t, h)} \left( \mathbf{c}^\top \mathbf{x}^t - \mathbf{c}^\top \mathbf{x}' \right) \geq 0, \tag{3}$$

and set

$$E(\mathcal{I}, \mathbf{x}^t, h) = \begin{cases} \dfrac{1}{\Delta^\star(\mathcal{I}, \mathbf{x}^t, h)}, & \Delta^\star(\mathcal{I}, \mathbf{x}^t, h) > 0, \\ +\infty, & \Delta^\star(\mathcal{I}, \mathbf{x}^t, h) = 0. \end{cases} \tag{4}$$

Let $\mathrm{LB}(k) = \{h \in \{0, 1\}^n : \|h\|_1 = k\}$ be the local-branching neighborhood family of size $k$. A Boltzmann distribution over neighborhoods of size $k^t$ is then

$$P(\mathcal{I}, \mathbf{x}^t, h) = \frac{\exp\left( -E(\mathcal{I}, \mathbf{x}^t, h) \right)}{\displaystyle\sum_{h' \in \mathrm{LB}(k^t)} \exp\left( -E(\mathcal{I}, \mathbf{x}^t, h') \right)}. \tag{5}$$

Because $\mathbf{x}^t \in \mathcal{M}(\mathcal{I}, \mathbf{x}^t, h)$ for any $h$, we have $\Delta^\star(\mathcal{I}, \mathbf{x}^t, h) \geq 0$. If a neighborhood is *ineffective* (no improvement over $\mathbf{x}^t$), then $\Delta^\star = 0$, hence $E = +\infty$ and $P(\mathcal{I}, \mathbf{x}^t, h) = 0$. Conversely, neighborhoods admitting larger improvements have smaller energy and thus higher probability.

**Independent (factorized) modeling.** Prior work (Han et al., 2023; Huang et al., 2023; Wu et al., 2021; Nair et al., 2020b) typically avoids modeling the full joint distribution over $h$ due to the prohibitive sampling cost in high dimensions, and instead assumes conditional independence across variables. Concretely, a message-passing GNN (MP-GNN) outputs

$$F_\theta(\mathcal{I}, \mathbf{x}^t) = (\hat{p}_1, \dots, \hat{p}_n), \quad \hat{p}_j \approx P(h_j = 1 \mid \mathcal{I}, \mathbf{x}^t),$$

and the joint is factorized as

$$\hat{P}(\mathcal{I}, \mathbf{x}^t, h) \approx \prod_{j=1}^n \hat{p}_j^{h_j} (1 - \hat{p}_j)^{1 - h_j}. \tag{6}$$

Under this independence assumption, the maximum-probability neighborhood of size $k^t$ is obtained by selecting the $k^t$ variables with the largest $\hat{p}_j$ (i.e., a top-$k^t$ rule), which explicitly biases the search toward variables with high marginal destruction probability while ignoring inter-variable coupling.

## 4 INDEPENDENCE ASSUMPTION FAILS IN REPRESENTING THE PROBABILITY DISTRIBUTION

The number of neighborhoods of size $k^t$ is $\binom{n}{k^t}$, which grows combinatorially (and exponentially in $n$ when $k^t = \Theta(n)$). To keep LNS applicable on large-scale problems, most methods factorize neighborhood selection into independent per-variable decisions. In this subsection, we present a counterexample showing that the independence assumption can assign *equal* estimated probability to two neighborhoods whose *true* utilities (and thus probabilities under equation 5) differ.

**Theorem 4.1** *There exists an ILP instance $\mathcal{I}$ with incumbent solution $\mathbf{x}^t$, and two neighborhoods with indicators $h^{(1)}$ and $h^{(2)}$, such that for any MP-GNN producing per-variable marginals $\hat{p}_j \approx P(h_j = 1 \mid \mathcal{I}, \mathbf{x}^t)$, the factorized estimate*

$$\hat{P}(\mathcal{I}, \mathbf{x}^t, h) = \prod_{j=1}^{n} \hat{p}_j^{h_j} (1 - \hat{p}_j)^{1 - h_j}$$

*satisfies $\hat{P}(\mathcal{I}, \mathbf{x}^t, h^{(1)}) = \hat{P}(\mathcal{I}, \mathbf{x}^t, h^{(2)})$, while the true probability defined in equation 5 satisfies $P(\mathcal{I}, \mathbf{x}^t, h^{(1)}) \neq P(\mathcal{I}, \mathbf{x}^t, h^{(2)})$.*

*Proof of theorem* 4.1: Consider $n = 8$ and incumbent $\mathbf{x}^t = (1, 0, 1, 0, 1, 0, 1, 0)$. Assume the ILP is represented as a bipartite graph whose 1-WL refinement yields two equivalence classes by parity, so that variables with the same parity share the same WL label (denote the label by $\chi$): $\chi(x_i) = \chi(x_j)$ iff $(i - j) \bmod 2 = 0$. By standard results on MP-GNN expressivity (cf. Theorem B.3), a 1-WL-bounded MP-GNN must output identical scores within each class; hence, for some $\alpha, \beta \in (0, 1)$,

$$\hat{p}_1 = \hat{p}_3 = \hat{p}_5 = \hat{p}_7 = \alpha, \qquad \hat{p}_2 = \hat{p}_4 = \hat{p}_6 = \hat{p}_8 = \beta.$$

Let $k^t = 4$ and define

$$h^{(1)} = (1, 1, 1, 1, 0, 0, 0, 0), \qquad h^{(2)} = (1, 1, 1, 0, 0, 0, 0, 1).$$

Both neighborhoods free two odd-parity variables and two even-parity variables, so their factorized estimates coincide:

$$\hat{P}(\mathcal{I}, \mathbf{x}^t, h^{(1)}) = \alpha^2 \beta^2 (1 - \alpha)^2 (1 - \beta)^2 = \hat{P}(\mathcal{I}, \mathbf{x}^t, h^{(2)}).$$

Now construct $\mathcal{I}$ so that an improvement is possible *only* when a specific coupled subset of variables is freed together (e.g., a pairwise-coupling constraint allowing a simultaneous flip of $(x_1, x_3)$ but not individually, with additional tying constraints involving $x_4$). Choose the couplings so that freeing the set indicated by $h^{(1)}$ enables a strictly better feasible solution $\mathbf{x}' \in \mathcal{M}(\mathcal{I}, \mathbf{x}^t, h^{(1)})$ with $\mathbf{c}^\top \mathbf{x}' < \mathbf{c}^\top \mathbf{x}^t$, whereas $\mathcal{M}(\mathcal{I}, \mathbf{x}^t, h^{(2)})$ contains no improvement (i.e., it is ineffective and only contains $\mathbf{x}^t$). Then

$$\Delta^\star(\mathcal{I}, \mathbf{x}^t, h^{(1)}) > 0 \quad \text{but} \quad \Delta^\star(\mathcal{I}, \mathbf{x}^t, h^{(2)}) = 0,$$

which, by equation 4 and equation 5, implies

$$P(\mathcal{I}, \mathbf{x}^t, h^{(1)}) > 0 \quad \text{and} \quad P(\mathcal{I}, \mathbf{x}^t, h^{(2)}) = 0,$$

hence $P(\mathcal{I}, \mathbf{x}^t, h^{(1)}) \neq P(\mathcal{I}, \mathbf{x}^t, h^{(2)})$ while the factorized estimates are equal.

The mismatch arises because the independence assumption collapses neighborhood quality to a product of marginal scores and is blind to feasibility/optimality effects that are triggered only by *jointly* freeing a coupled set of variables. Consequently, two neighborhoods that look identical under factorized estimates can have sharply different true utilities.

$$\begin{aligned}
\min \quad & 2x_1 + x_2 + 2x_3 + x_4 + 2x_5 + x_6 + 2x_7 + x_8, \\
\text{s.t.} \quad & x_1 + x_2 = 1, \; x_2 + x_3 = 1, \; x_3 + x_4 = 1, \; x_4 + x_1 = 1, \\
& x_5 + x_6 = 1, \; x_6 + x_7 = 1, \; x_7 + x_8 = 1, \; x_8 + x_5 = 1, \\
& x_j \in \{0, 1\}, \; \forall j \in \{1, 2, \dots, 8\}.
\end{aligned}$$

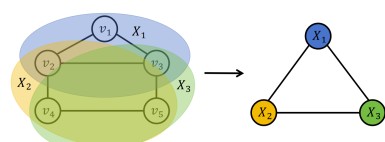

Figure 1: An ILP instance where two WL-equivalent neighborhoods lead to different effects.

Figure 2: Decomposition graph.

For the instance above, ML-based LNS under the independence assumption may struggle to predict an effective neighborhood. Each variable is judged only by its own marginal probability, ignoring inter-variable coupling that is critical for constructing effective neighborhoods. This shows that neglecting coupling can yield indistinguishable neighborhood estimates with markedly different true utilities, undermining LNS guided by such factorized models.

## 5 VARIABLE COUPLING ENHANCEMENT FOR LARGE NEIGHBORHOOD SEARCH

In the previous section, we reveal that ML-based LNS under the independence assumption might fail to account for variable coupling, leading to indistinguishable neighborhood predictions and reduced effectiveness. The current approaches include: adding random features as variable extra feature(Chen et al., 2023) and 2-FGNN framework(Chen et al., 2024c). However, as it has been validated in subsequent experiments, although adding random features can improve the GNN's fitting ability during the training phase, since random features do not contain meaningful ILP information, therefore they cannot effectively improve the GNN's generalization performance on the test set, potentially leading to overfitting. Since 2-FGNNs operate on each variable-constraint pair rather than on single variables, causing their message-passing state space to grow quadratically and their updates to involve cubic-time aggregation, which makes both memory $\mathcal{O}(n^2)$ and compute $\mathcal{O}(n^3)$ far more expensive. Therefore, for LNS that requires making rapid decisions on neighborhood selection, such expensive complexity is unacceptable for LNS on large-scale ILPs.

It is worth noting that the limitations of representing ILP as a bipartite graph arise from the insufficient expressive power of GNNs on bipartite graphs. No information is lost in the process of representing an ILP as a bipartite graph.

### 5.1 DECOMPOSITION GRAPH FOR VARIABLES

Given an ILP instances $\mathcal{I}$, the variable graph of $\mathcal{I}$ as $\mathcal{G}_n = (V, E)$, where its vertices set $V$ is the set of variables, two variables vertices $v_{i_1}, v_{i_2}$ are considered adjacent if the two variables appear in the same constraint. Graph decomposition refers to the process of breaking a graph $G = (V, E)$ into smaller subgraphs or vertex subsets as "bags" or "components", and optionally imposing a structure on these subgraphs. Formally, a graph decomposition is defined as follow, :

**Definition 5.1** *A graph decomposition of a graph $\mathcal{G} = (V, E)$, denoted as $Dec(\mathcal{G}) = (X, \Xi)$, is a collection of subgraphs or vertex subsets $X = \{X_1, X_2, \cdots X_k\}(X_i \subseteq V)$ (called bags or components) which serve as the vertices of the decomposition graph, subject to the following conditions, and :(1) Vertex coverage: Every vertex of $\mathcal{G}$ appears in at least one bag: $\bigcup_{i=1}^{k} X_i = V$.(2) Edge coverage: Each edge $(v_1, v_2) \in E$ is included in at least one bag $X_i \subseteq V$.(3) Structural relationship: In $Dec(\mathcal{G}) = (X, \Xi)$, two bags $X_i, X_j$ are considered adjacent if they share common variable nodes $(X_i \cap X_j \neq \emptyset \rightarrow (X_i, X_j) \in \Xi)$.*

Introducing graph decomposition into LNS aims to address the issue4: Destroy strategies based solely on variable independence tend to select strongly coupled variables that are split across the boundary, leading to uncoordinated updates. Moreover, if the destroy set severs strong couplings, the repair subproblem behaves as if it were trapped behind hard constraint walls, making improvements unlikely and time-consuming. As for graph decomposition, strongly coupled variables tend to cohabit the same bag (or neighboring bags) after chordal completion. In this LNS is able to Align destroy sets with bags keeps strongly coupled variables co-updated.

**Theorem 5.2** *Given an ILP instances $\mathcal{I}$, $\mathcal{G}_n = (V, E)$ is $\mathcal{I}$'s variable graph and $m$ denotes the number of constrains in $\mathcal{I}$, if the $i - th$ bag $X_i$ of vertex subsets $X = \{X_1, X_2, \cdots X_k\}(X_i \subseteq V)$ is formed by all the variable nodes involved in the $i - th$ corresponding constraint in $\mathcal{I}$, then it yields a graph decomposition of a graph $\mathcal{G}_n = (V, E)$.*

Theorem 5.2 provides a perspective: Since the set of constraints forms a graph decomposition, the representation of constraint nodes in the bipartite graph can be exploited to capture features characterizing the couplings among variables. In this way, the representation learning of the bipartite graph and the decomposition graph can be organically integrated to avoid extra computational for additional construction of a decomposition graph.

### 5.2 COUPLING-ENHANCED LARGE NEIGHBORHOOD SEARCH

As Figure 3 demonstrated, the process of coupling-enhanced large neighborhood search comprises following three stages:

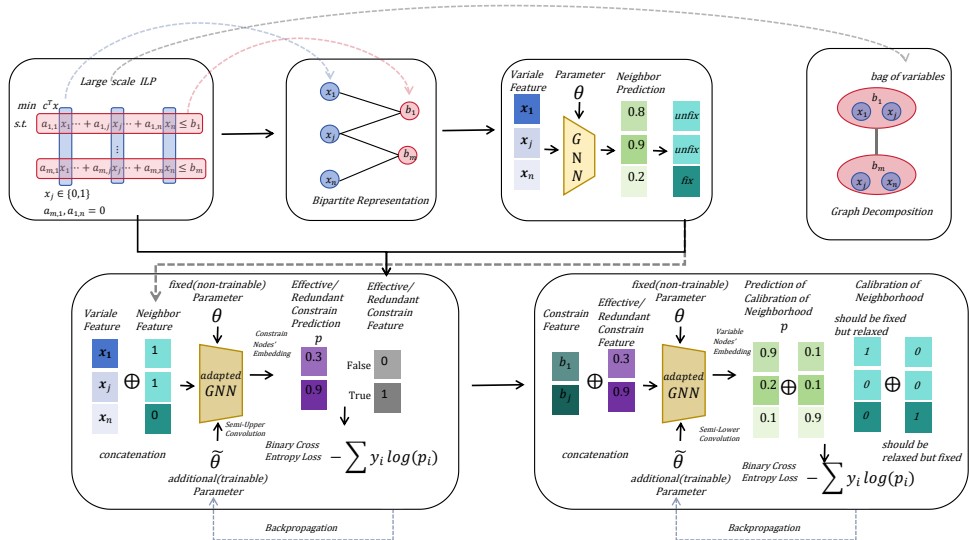

Figure 3: Framework of Coupling-Enhanced Large Neighborhood Search

**Contrastive Learning for LNS via GNN:** Following prior work on learning for ILPs(Huang et al., 2023; Gasse et al., 2019; Sonnerat et al., 2021; Wu et al., 2021), the policy network adopts a bipartite graph representation of the ILP. Each node and edge is enriched with features, including those from Gasse et al. (2019); Huang et al. (2023); Sonnerat et al. (2021) to provide a richer representation. The LNS policy is implemented as a GNN with learnable parameters, which first maps features into a latent space via embedding layers and then performs two rounds of message passing:(1) constraint nodes attend to their neighboring variables (2) variable nodes attend to their neighboring constraints. The resulting variable embeddings are passed through a multilayer perceptron followed by a activation function to produce variable-wise selection scores $\hat{h}^t \in [0,1]^n$. Training is conducted using a supervised contrastive loss(InfoNCE), which encourages the predicted actions to align closely with positive samples and diverge from negative ones. At inference time, the learned policy is integrated into the LNS framework: in each iteration, the $k_t$ variables with the highest scores are greedily selected for reoptimization to generated the neighborhood selection, the indicator is denoted as $\hat{h}^t \in \{0,1\}^n$.

**Alignment for Bags of Variables:** Due to the convenience of regarding constraint as variable bag by theorem 5.2, we align the features of variables' bag with the constraint features obtained after the neighborhood selection, discrepancy between the predicted and real-world features can be regarded as the predictive loss caused by the independence assumption in the first-stage LNS that ignores variable couplings to feedback the coupling effects among variables. It refers to fine-tuning the first layer of message passing in the GNN that has already been trained during the first stage. Specifically, the neighborhood selection indicator $\hat{h}^t \in \{0,1\}^n$ is added as variable's extra feature, meanwhile trainable parameter connected in parallel with the originally learned weights is added into the first-layer message passing in corresponding to the feature vector $\hat{h}^t \in \{0,1\}^n$, and the learned weights are fixed during the training. After the first message passing, the embeddings of the constraints are further processed by two additional MLPs in order to output probabilistic predictions $\hat{r} \in [0,1]^{d \times m}$($d$ refers the feature dimension of constraints) over the constraint features obtained after the neighborhood selection, which includes constraint's effectiveness and redundancy.

**Calibration of Neighborhood Selection:** In the second stage, the discrepancy between the predicted and real-world features of the bags(constraints) is incorporated as an additional input feature for the second-layer message passing. The features of training samples $h_{des}^{t*} \in \{0,1\}^n$ and $h_{fix}^{t*} \in \{0,1\}^n$ for stage III are generated by the prediction indicator $\hat{h}$ and real-world neighborhood indicator $h^t$, where $h_{des}^{t*}$ refers to whether variables are relaxed but should be destroyed, and $h_{fix}^{t*}$ refers to whether variables are fixed but should be relaxed. Similar to the second stage, we

apply fine-tuning to the second layer of message passing, as adding trainable parameter connected in parallel with the originally learned weights in corresponding to the extra feature and fixed the learned weights during the training. After through a two additional MLPs, the framework outputs calibration scores $(\hat{h}_{des}^t, \hat{h}_{fix}^t) \in [0,1]^n$ as the prediction of $(h_{des}^t, h_{fix}^t)$. The framework Greedily selects $\beta_1 \cdot k^t \cdot n_t$ ($n_t$ denotes the proportion $\hat{h}^t$ and $h^{t-1}$ overlaps, parameter $\beta_1 < 1$) highest-scoring of both $\hat{h}_{des}^t$ and $\hat{h}_{fix}^t$ variables with equal number to transform their states(relaxed/fixed).

**Remark:** (1)The samples in the stage III are generated by combining the predictions $\hat{h}^t \in \{0,1\}^n$ from the first stage with the samples, and therefore, compared to Huang et al. (2023), our framework does not require additional samples. (2) Stage III encourages neighborhood to be different than previous, since the higher proportion $\hat{h}^t$ and $h^{t-1}$ overlaps, the more states of variables will be transformed. In Appendix $E$, we provide the full details of fine-tuning two graph network architecture:graph convolutional network(GCN) and graph attention network(GAT).

Denote the collection of all GNNs under our coupling-enhanced framework as $\mathcal{F}_{CE_{GNN}}$, then we have the following theorem:

**Theorem 5.3** *Given any ILP instance $\mathcal{I}$, $\mathbf{x}^t \in \{0,1\}^n$ is the incumbent solution, $h^t \in \{0,1\}^n$ as neighborhood selection indicator generated from first stage and $\mathcal{M}(\mathcal{I}, \mathbf{x}, h^t)$ as the neighbors searching space, then $\forall \epsilon, \phi > 0$, there is $F \in \mathcal{F}_{CE_{GNN}}$ such that the following holds:*

- *For effective and redundant indicator $\mathcal{R}(\mathcal{I}, \mathbf{x}^t, h), \mathcal{T}(\mathcal{I}, \mathbf{x}^t, h)$, we have*

$$P(\|\hat{\mathcal{R}} - \mathcal{R}(\mathcal{I}, \mathbf{x}^t, h)\| > \phi) < \epsilon, \quad P(\|\hat{\mathcal{T}} - \mathcal{T}(\mathcal{I}, \mathbf{x}^t, h)\| > \phi) < \epsilon$$

  *where $\hat{\mathcal{R}}, \hat{\mathcal{T}}$ are probabilistic predictions for effective and redundant indicator generated from the second stage.*

- *Denote $\tilde{h}^t \in [0,1]^n$ as the output from $F$ after calibration, if the scale of neighborhood $k^t < \frac{n}{2}$, then we have*

$$P(\|\tilde{h}^t - h_0^t\| > \phi) < \epsilon$$

  *where $h_0^t$ is optimal neighborhood selection indicator:$\max_{h \in LB(k^t)} \left( E(\mathcal{I}, \mathbf{x}^t, h) \right)$ ($E(\mathcal{I}, \mathbf{x}^t, h), LB(k)$ is defined in equation ??)*

# 6 EMPIRICAL EVALUATION

In this section, we introduce our evaluation setup and then present the results.

**Benchmark Dataset**(1).*Generated Instance:* We evaluate on four NP-hard problem benchmarks that are widely used in existing studies Zhang & Others (2024); Wu et al. (2021); Song et al. (2020); Scavuzzo et al. (2022): the minimum vertex cover (MVC), maximum independent set (MIS) problems, combinatorial auction(CA) and set covering(SC) problems. 100 instances for each problem are generated as a test set. We first generate a test set of 100 instances for each problem, namely MVC-S, MIS-S, CA-S and SC-S. For each test set, Table 2 shows its average numbers of variables and constraints. (2)*Real-World:* We evaluate on two benchmarks, IP and WA, come from two challenging real-world problem families used in NeurIPS ML4CO 2021 competition (Gasse et al., 2022). We use 240 training, 60 validation, and 100 testing instances, following the settings in Han et al. (2023). Please refer to Appendix $F$ for more details on the benchmarks.

**Baselines:** (1) BnB using SCIP (v8.0.1), the state-of-the-art open-source ILP solver;(2)Random: LNS which selects the neighborhood by uniformly sampling $k^t$ variables without replacement;(3)GCN-CL LNS: Graph Convolutional Network with Contrastive Learning (Huang et al., 2023; Kipf & Welling, 2017) (4)GCN-CE LNS:Graph Convolutional Network with our framework (Kipf & Welling, 2017) (5)GAT-CL LNS: Graph Attention Network with Contrastive Learning (Huang et al., 2023; Brody et al., 2021) (6)Graph Attention Network with our framework (Brody et al., 2021). Further details of instance generation are included in Appendix.

**Metrics:** We use the following metrics to evaluate all baselines approaches:(1) The *primal gap* Berthold (2006) is the normalized difference between the primal bound $v$ and a precomputed best

Table 1: Primal gap (PG) (in percent), primal integral (PI) at 60 minutes runtime cutoff, averaged over 100 test instances and their standard deviations for generated instances. "↓" means the lower the better.

| | PG (%) ↓ | PI ↓ | PG (%) ↓ | PI ↓ | PG (%) ↓ | PI ↓ | PG (%) ↓ | PI ↓ |
|---|---|---|---|---|---|---|---|---|
| | MVC-S | | MIS-S | | CA-S | | SC-S | |
| BnB | 1.41±0.25 | 61.1±17.2 | 4.87±1.29 | 199.7±78.6 | 2.31±0.66 | 144.8±33.7 | 1.21±0.75 | 81.1±39.1 |
| RANDOM | 1.20±0.95 | 51.1±36.1 | 0.31±0.12 | 23.1±7.8 | 6.1±1.31 | 267.7±44.1 | 2.71±1.33 | 137.1±33.1 |
| CL-GCN | 0.31±0.26 | 15.52±11.90 | 0.27±0.15 | 22.04±9.65 | 1.16±0.90 | 89.15±40.81 | 0.50±1.03 | 46.15±21.92 |
| CE-GCN | 0.20±0.19 | 12.06±9.50 | 0.18±0.12 | 14.95±6.79 | 0.86±0.70 | 69.16±28.33 | **0.40±0.74** | **26.80±18.56** |
| CL-GAN | 0.19±0.11 | 10.46±7.85 | 0.19±0.19 | 16.50±6.30 | 0.78±0.36 | 63.32±26.59 | 0.43±0.66 | 30.93±17.70 |
| CE-GAN | **0.14±0.13** | **6.24±5.18** | **0.13±0.08** | **10.44±4.43** | **0.47±0.42** | **39.85±21.69** | 0.50±0.32 | 46.99±10.69 |
| | MVC-L | | MIS-L | | CA-L | | SC-L | |
| BnB | 2.63±0.40 | 133.6±11.5 | 6.16±1.74 | 275.8±19.6 | 2.62±1.98 | 344.3±90.5 | 1.69±0.97 | 111.1±42.9 |
| RANDOM | 0.37±0.25 | 23.4±8.6 | 0.19±0.11 | 19.7±7.2 | 5.32±0.81 | 237.0±24.2 | 3.18±1.82 | 179.1±58.4 |
| CL-GCN | 0.25±0.08 | 21.7±7.6 | 0.27±0.24 | 29.2±13.5 | 0.21±0.09 | 262.3±39.1 | 1.27±1.01 | 81.9±50.8 |
| CE-GCN | 0.21±0.17 | 17.9±13.6 | 0.23±0.15 | 24.8±7.7 | 0.20±0.15 | 256.9±64.0 | 1.29±0.46 | 82.0±22.7 |
| CL-GAN | 0.08±0.05 | 10.3±4.3 | 0.15±0.14 | 16.4±8.5 | 0.12±0.08 | 148.5±32.8 | 0.72±0.27 | 47.8±7.1 |
| CE-GAN | **0.07±0.09** | **8.8±8.7** | **0.13±0.07** | **16.2±5.3** | **0.10±0.05** | **116.3±19.6** | **0.63±0.47** | **41.2±27.0** |

known objective value $v^*$, defined as $\frac{|v-v^*|}{\max(v,v^*,\epsilon)}$ if $v$ exists and $v \cdot v^* \geq 0$, or 1 otherwise. We use $\epsilon = 10^{-8}$ to avoid division by zero;(2) The *primal integral* Achterberg et al. (2012) at time $q$ is the integral on $[0, q]$ of the primal gap as a function of runtime. It captures the quality of and the speed at which solutions are found;

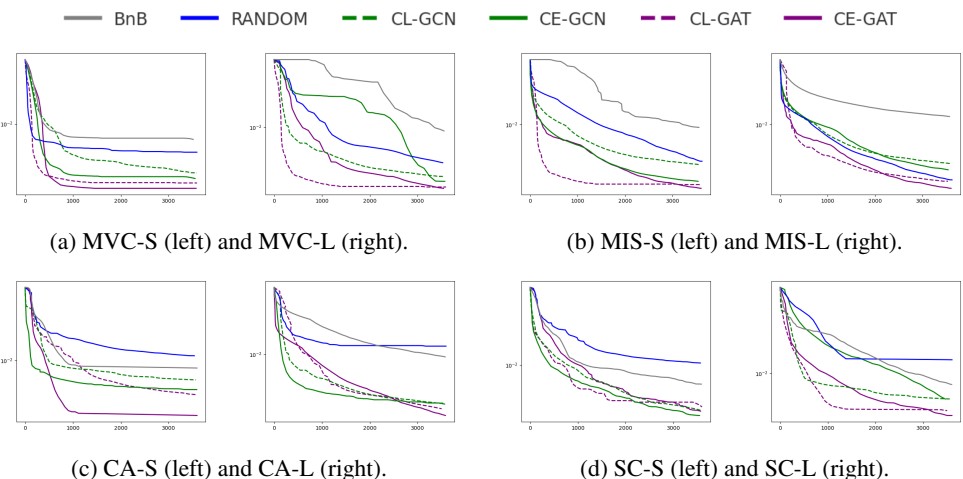

(a) MVC-S (left) and MVC-L (right).  (b) MIS-S (left) and MIS-L (right).

(c) CA-S (left) and CA-L (right).  (d) SC-S (left) and SC-L (right).

Figure 4: Primal gap (PG) (in percent), primal integral of Generated Instance.

## 6.1 RESULTS

Figure 4 shows the primal gap as a function of runtime. Table 5 presents the average primal gap and primal integral at 60 minutes runtime cutoff on small and large instances, respectively. (Appendix $G$ provides the results at, 30 and 45 minutes runtime cutoff ). The result shows significantly better anytime performance than the original contrastive learning baselines(Huang et al., 2023) on all problems, achieving the smallest average primal gap and primal integral. Figure 5 shows that our framework demonstrates significant advantages over the baseline models not only on generated instances but also on real-world instances. In Appendix, we present strong results in comparison with two more baselines and on one more performance metric.

## 6.2 ABLATION STUDY

We evaluate how alignment for bags of variables in stage II and calibration of neighborhood selection in stage III contribute to our framework's performance. (1) For the alignment, we replace the second-stage constraint features with the random features from Chen et al. (2024b). and (2) We changed the

| | PG (%) ↓ | PI ↓ | PG (%) ↓ | PI ↓ |
|---|---|---|---|---|
| | WA | | IP | |
| BnB | 0.47±0.11 | 3.41 ±0.81 | 30.6±3.5 | 7.35±0.57 |
| RANDOM | 0.35±0.09 | 2.91 ±0.72 | 37.1±4.5 | 8.91±0.72 |
| CL-GCN | 0.41±0.16 | 3.15±1.13 | 27.3±3.1 | 6.81±0.51 |
| CE-GCN | 0.17±0.07 | 1.43±0.43 | 22.1±4.2 | 5.29±0.67 |
| CL-GAN | 0.26±0.19 | 2.12±1.51 | 28.1±2.2 | 6.97±0.35 |
| CE-GAN | **0.12±0.09** | **1.07±0.61** | **21.5±3.7** | **5.14±0.72** |

Figure 5: Primal gap (PG) (in percent), primal integral (PI) at 60 minutes runtime cutoff for real-world instances

value of the coefficient $\beta_1$ from 0.2(denoted as CE-GCN/GAT-0.2) to 0.4(denoted as CE-GCN/GAT-0.4) to test its impact on the experiments. Figure 6 shows the primal gap for the first ablation study. The difference in the primal gap between the random method and our method demonstrates the necessity of alignment for bags of variables. Figure 7 presents the primal gap for the second ablation study. The result shows if $\beta_1$ is set too large, it will affect the performance in the early stage, whereas if it is too small, there will be limited room for optimization in the subsequent iterations.

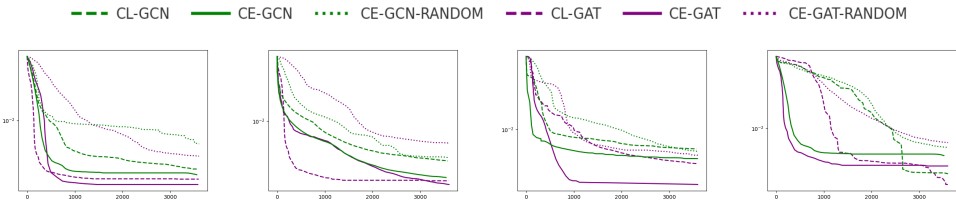

Figure 6: The primal gap for first ablation and the dataset, from left to right are (1)MVC-S (2)MIS-S (3)CA-S (4)WA, respectively

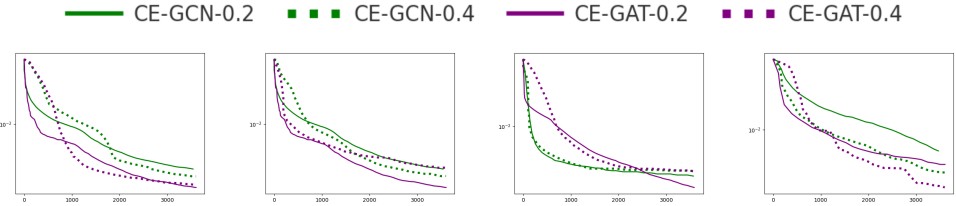

Figure 7: The primal gap for second ablation and the dataset, from left to right are (1)MVC-L (2)MIS-L (3)CA-L (4)SC-L, respectively

## 7 CONCLUSION

This paper introduces Coupling-Enhanced Neural Large Neighborhood Search (CE-LNS) for solving integer linear programs (ILPs). Neural LNS frameworks assume variable independence, which overlooks coupling, leading to indistinguishable predictions. To address this, CE-LNS augments GNN-based neighborhood prediction with graph decomposition. This refinement allows the model to capture coupling relationships between variables and calibrate neighborhood selection accordingly. Theoretically, CE-LNS can identify effective vs. redundant constraints and approximate optimal neighborhoods. Empirically, it consistently outperforms existing neural LNS methods across synthetic and real-world ILP benchmarks, achieving smaller primal gaps, better anytime performance, and stronger generalization.

## 8 ETHICS STATEMENT

We acknowledge the ICLR Code of Ethics and confirm that our work adheres to its principles. Our research prioritizes societal benefit, avoids harm, and respects privacy and intellectual property. All data used in this study comply with ethical guidelines and relevant licenses.

## 9 REPRODUCIBILITY STATEMENT

To ensure the reproducibility of our work, we have made substantial efforts to provide comprehensive details and resources across our main paper, appendix, and supplementary materials.

*Code and Resources.* We have developed a reproducible codebase MTG, extended to support our message tuning. Our code is available at https://anonymous.4open.science/r/CE-LNS-3stage-05EA/. Anonymous, downloadable source code also includes scripts for pre-training, adaptation, and evaluation on all datasets used in our experiments.

*Theoretical Proofs.* All theoretical claims are rigorously proven in Appendix $C$ and $D$.

We believe these efforts collectively ensure the reproducibility of our work and encourage the community to build upon our findings.

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

## THE USE OF LARGE LANGUAGE MODELS (LLMs)

In the preparation of this manuscript, we employed large language models (LLMs) solely for the purpose of language polishing and refinement. Specifically, LLM-assisted editing was used to improve grammatical accuracy, sentence fluency, and terminological consistency across the paper, particularly in sections where non-native expressions might affect readability. All substantive intellectual contributions—including the formulation of research ideas, theoretical analysis, algorithm design, experimental setup, result interpretation, and conclusions—remain entirely our own.

## A    ADDITIONAL RELATED WORK

Large Neighborhood Search (LNS) has a long history as a primal heuristic inside branch-and-bound (BnB) solvers. In that setting, LNS-based primal heuristics are invoked periodically at selected nodes of the search tree, and their activation schedule is dynamic because LNS calls are typically more expensive than many other primal heuristics. Compared with an external (stand-alone) LNS for ILPs-whose goal is likewise to improve the incumbent but operates outside the BnB control flow-two differences are central: (i) BnB LNS is interleaved with tree search and triggered at varying depths and times; (ii) its destroy (neighborhood-defining) mechanisms often leverage node-local information such as the node's LP relaxation and dual bound, which are not directly available or portable in a stand-alone LNS setting.

**LNS Heuristics for ILPs:** The Crossover heuristic (Rothberg, 2007) destroys variables whose values differ across a small set of selected solutions (typically two), whereas the Mutation heuristic (Rothberg, 2007) destroys a random subset of variables. The RINS heuristic (Danna et al., 2005) destroys variables whose values differ between the node's LP-relaxation solution and the incumbent solution. Relaxation Enforced Neighborhood Search (RENS) (Berthold, 2014) restricts the neighborhood to be the feasible roundings of the LP relaxation at the current search tree node. Local Branching (LB)(Fischetti & Lodi, 2003) restricts the neighborhood to a ball around the current incumbent solution. Distance Induced Neighborhood Search (DINS) (Ghosh, 2007) takes the intersection of the neighborhoods of the Crossover, Local Branching and Relaxation Induced Neighborhood Search heuristics.Graph-Induced Neighborhood Search (GINS) (Maher et al., 2017) destroys the breadth-first-search neighborhood of a variable in the bipartite graph representation of the ILP.

**Neural Method for LNS:** Wu et al. (2021) train a Neural Diving model and an imitation-learned neighborhood selector to drive LNS on mixed-integer programs with a MIP solver in the loop. Sonnerat et al. (2021) use deep reinforcement learning to learn which variables to destroy/repair, turning LNS for integer programming into an effective learned policy. Huang et al. (2023) propose that CL-LNS learns a contrastive destroy heuristic on ILP bipartite graphs, yielding strong anytime performance on standard ILP benchmarks. Hottung & Tierney (2019) propose a neural repair operator with attention inside an LNS for CVRP/SDVRP, markedly improving over handcrafted LNS. Johnn et al. (2024) propose Graph-RL framework picks ALNS operators conditioned on the current solution state to outperform classic adaptive layers.Wu et al. (2021) formulates large neighborhood search as a reinforcement learning problem where a policy network learns to select which variables to destroy and repair, achieving better anytime performance on integer programming than classical heuristics. Nair et al. (2020a) implement for the Neural Neighborhood Selection approach used in learning-based LNS for MIPs. Liu et al. (2025) proposes APOLLO-ILP, an alternating prediction–correction neural framework that integrates learnable heuristics with exact solvers to improve both efficiency and solution quality for ILP. **?** proposes a hybrid framework that leverages Graph Neural Networks (GNNs) and Gradient Boosted Decision Trees (GBDTs) to accelerate large-scale integer programming optimization. **?** introduces a lightweight optimizer that achieves efficient solutions for large-scale ILPs using only a small-scale training dataset.

**Graph-Based Expressive power of GNNs for ILPs :** Xu et al. (2019) proves that message-passing GNNs are at most as expressive as the 1-Weisfeiler–Lehman test and introduces the Graph Isomorphism Network (GIN), which matches 1-WL's power and empirically achieves state-of-the-art performance on graph classification tasks. Chen et al. (2023) formalizes ILPs in GNN terms and shows 1-WL–bounded MPNNs have intrinsic limits (foldable cases) while still approximating key properties for non-foldable instances. Chen et al. (2022) proves GNNs (and the 1-WL test) have sufficient separation power to distinguish LPs under a principled encoding, clarifying when message

passing suffices. Chen et al. (2024b) characterizes when MPNNs can represent feasibility/optima for QPs and MIQPs, pinpointing WL-style expressivity boundaries and practically testable criteria. Chen et al. (2024c) studies whether MPNNs can emulate strong branching, revealing structural/expressivity hurdles and conditions under which approximation is viable. Chen et al. (2025) shows permutation symmetries in ILPs confound 1-WL–level GNNs and proposes orbit-based augmentation to break WL-indistinguishability among symmetric variables. Chen et al. (2024a) introduces a symmetry-aware ILP learning pipeline that mitigates the WL/MPNN inability to separate symmetric variables in standard encodings. Gasse et al. (2019) pioneers the variable–constraint bipartite ILP encoding for GNN-guided branching, implicitly operating at 1-WL expressivity and motivating later expressivity analyses. Gupta et al. (2020) combines a GNN root-encoding with light node-specific models, illustrating practical gains while reflecting 1-WL–bounded representation on ILP graphs. Paulus et al. (2022) uses (tri/bipartite) graph encodings for cut selection, highlighting how representational choices interact with WL-style expressivity in ILP states. Khalil et al. (2022) General bipartite-graph GNN framework for ILP guidance, exemplifying strengths and 1-WL–type limits on variable–constraint interaction encoding. Labassi et al. (2022) Siamese GNNs compare B&B nodes via bipartite encodings, implicitly constrained by WL equivalence classes of solver states.

# B PRELIMINARIES

## B.1 WEISFEILER–LEHMAN TEST FOR ILPS

The 1-dimensional Weisfeiler–Lehman algorithm (1-WL), also known as *color refinement*, iteratively computes a color map $\chi_G$ for a graph $G = (V, E)$, assigning each vertex $v \in V$ a color $\chi_G(v) \in \mathcal{C}$. We apply 1-WL to the weighted bipartite ILP graph described below. At initialization, vertices receive colors derived from node features (e.g., $c_j$ on variables and $b_i$ on constraints). At each iteration, the color of a vertex is updated by hashing its previous color together with a multiset of its neighbors' colors paired with incident edge weights $A_{i,j}$. This refinement repeats for $L$ iterations or until stabilization.

---

**Algorithm 1** 1-WL on a weighted bipartite ILP graph

---

1: **Input:** ILP $\mathcal{I} = (\mathbf{A}, \mathbf{b}, \mathbf{c})$; bipartite graph $G = (V \cup U, E)$ with constraint nodes $V = \{v_i\}_{i=1}^m$, variable nodes $U = \{u_j\}_{j=1}^n$; iterations $L$.
2: **Init:** $\chi^0(v_i) \leftarrow \text{hash}(b_i)$, $\chi^0(u_j) \leftarrow \text{hash}(c_j)$.
3: **for** $\ell = 1$ **to** $L$ **do**
4:    **for each** $v_i \in V$ **do**
5:       $\chi^\ell(v_i) \leftarrow \text{hash}\big(\chi^{\ell-1}(v_i), \{\{(\chi^{\ell-1}(u_j), A_{i,j}) : u_j \in N(v_i)\}\}\big)$
6:    **end for**
7:    **for each** $u_j \in U$ **do**
8:       $\chi^\ell(u_j) \leftarrow \text{hash}\big(\chi^{\ell-1}(u_j), \{\{(\chi^\ell(v_i), A_{i,j}) : v_i \in N(u_j)\}\}\big)$
9:    **end for**
10: **end for**
11: **Output:** stabilized labels $\chi(v_i), \chi(u_j)$.

---

Each iteration refines the partition induced by colors on $V \cup U$. Since $V \cup U$ is finite, the process stabilizes in at most $|V| + |U|$ refinements. The 1-WL procedure is a powerful heuristic for graph isomorphism: if the stabilized color multisets differ, the graphs are not isomorphic; if they coincide, the graphs may still be non-isomorphic (1-WL is a necessary but not sufficient test). These limitations motivate higher-order WL tests with stronger expressivity.

**Lemma B.1** *Let $n$ be the number of variables and $m$ the number of constraints. For an ILP instance $\mathcal{I}$, incumbent $\mathbf{x}^t$, and neighborhood indicator $h$:*

- *If the neighborhood $\mathcal{M}(\mathcal{I}, \mathbf{x}^t, h)$ is effective (i.e., admits an improving feasible solution), then at least one constraint is effective for this neighborhood.*

- *There exist ILP instances for which, relative to a given neighborhood, no constraint is re-dundant (i.e., removing any single constraint changes the feasible set or the best achievable objective within the neighborhood).*

**Theorem B.2 (Weighted isomorphism preserves ILP solutions)** *Let $\mathcal{I}_1 = (\mathbf{A}_1, \mathbf{b}_1, \mathbf{c}_1)$ and $\mathcal{I}_2 = (\mathbf{A}_2, \mathbf{b}_2, \mathbf{c}_2)$ be ILPs with weighted bipartite graphs $\mathcal{G}^1$ and $\mathcal{G}^2$. If there exists a weight-preserving bipartite isomorphism that simultaneously permutes constraint/variable indices and carries $(\mathbf{A}_1, \mathbf{b}_1, \mathbf{c}_1)$ to $(\mathbf{A}_2, \mathbf{b}_2, \mathbf{c}_2)$, then $\mathcal{I}_1$ and $\mathcal{I}_2$ have the same optimal objective value and isomorphic sets of optimal solutions.*

Following Xu et al. (2019), the separation power of message-passing GNNs (MP-GNNs) is upper-bounded by 1-WL: nodes indistinguishable by 1-WL will receive identical MP-GNN embeddings. Consequently, under the factorized action model equation 6, the MP-GNN-predicted marginal probabilities preserve inherent graph symmetries.

**Theorem B.3 (Xu et al. (2019))** *Let $\mathcal{I}$ be an ILP and consider its bipartite graph. For any MP-GNN and any two variable nodes $u_{j_1}, u_{j_2}$ with identical 1-WL labels, if $p(\mathcal{I}, x_j, h_j = 1)$ in equation 6 is represented by the MP-GNN output $\hat{p}_j$, then $\hat{p}_{j_1} = \hat{p}_{j_2}$.*

## B.2 Invariance and Equivariance for Bipartite ILP Graphs

An integer linear program

$$\min_{\mathbf{x} \in \{0,1\}^n} \mathbf{c}^\top \mathbf{x} \quad \text{s.t.} \quad \mathbf{A}\mathbf{x} \le \mathbf{b}$$

admits a weighted bipartite representation $G = (V \cup U, E)$ with constraint nodes $V = \{v_i\}_{i=1}^m$, variable nodes $U = \{u_j\}_{j=1}^n$, and edges only across $V$ and $U$. Node features are $b_i$ on $v_i$ and $c_j$ on $u_j$; edge weights store coefficients $A_{i,j}$. Let $S_m$ and $S_n$ be permutation groups acting on $V$ and $U$, respectively. For $(\sigma_V, \sigma_U) \in S_m \times S_n$, define $(\sigma_V, \sigma_U)G$ as the graph obtained by jointly permuting constraint/variable indices, their features, and rows/columns of $\mathbf{A}$ accordingly; this action leaves the optimization problem invariant.

We consider three canonical mappings of $G$:

1. *Feasibility $\Phi_{\text{feas}}(G) \in \{0, 1\}$.*

2. *Optimal objective $\Phi_{\text{obj}}(G) \in \mathbb{R} \cup \{\pm\infty\}$.*

3. *An optimal solution $\Phi_{\text{solu}}(G) \in \{0, 1\}^n$ (when a canonical choice is fixed).*

**Definition B.4 (Invariance and equivariance)** *A graph-level map $\Psi(G)$ is invariant to $(\sigma_V, \sigma_U)$ if $\Psi((\sigma_V, \sigma_U)\cdot G) = \Psi(G)$ for all $(\sigma_V, \sigma_U)$. A variable-level map $f(G) \in \mathbb{R}^n$ is equivariant if $f((\sigma_V, \sigma_U)\cdot G) = \sigma_U(f(G))$, i.e., the output permutes in the same way as variables.*

The following is standard (e.g., Chen et al. (2022)):

**Theorem B.5 (Symmetry properties of ILP quantities)** *For any $(\sigma_V, \sigma_U) \in S_m \times S_n$,*

$$\Phi_{\text{feas}}((\sigma_V, \sigma_U)\cdot G) = \Phi_{\text{feas}}(G), \quad \Phi_{\text{obj}}((\sigma_V, \sigma_U)\cdot G) = \Phi_{\text{obj}}(G),$$

*and*

$$\Phi_{\text{solu}}((\sigma_V, \sigma_U)\cdot G) = \sigma_U(\Phi_{\text{solu}}(G)).$$

*Thus, feasibility and optimal value are invariant, while an optimal solution is equivariant under variable permutations.*

## B.3 Universal Approximation for Permutation-Invariant Set Functions

Let $\mathcal{X} \subset \mathbb{R}^d$ be compact and $\mathcal{X}^{\le M} := \bigcup_{m=0}^M \mathcal{X}^m$ denote multisets of size at most $M$. A function $F$ on finite sets $S \subset \mathcal{X}$ is *permutation-invariant* if $F(S) = F(\pi S)$ for any permutation $\pi$ of the elements of $S$. We write $S = \{x_1, \ldots, x_m\}$ (order arbitrary), and $\sum_{x \in S}$ denotes multiset summation.

**Definition B.6 (Deep Sets)** *A map $F$ has* Deep-Set form *if there exist $\rho : \mathbb{R}^p \to \mathbb{R}^k$ and $\phi : \mathcal{X} \to \mathbb{R}^p$ such that*

$$F(S) = \rho\left(\sum_{x \in S} \phi(x)\right).$$

Chen et al. (2024b) prove a universal approximation theorem for invariant set functions:

**Theorem B.7 (Universal approximation of invariant set functions)** *Fix $M \in \mathbb{N}$. Let $F : \mathcal{X}^{\leq M} \to \mathbb{R}^k$ be continuous (with the topology that pads sets of size $m < M$ by a fixed dummy element). Then for every $\varepsilon > 0$ there exist continuous $\phi : \mathcal{X} \to \mathbb{R}^p$ and $\rho : \mathbb{R}^p \to \mathbb{R}^k$ such that*

$$\sup_{S \in \mathcal{X}^{\leq M}} \left\| F(S) - \rho\left(\sum_{x \in S} \phi(x)\right) \right\| < \varepsilon.$$

*Moreover, if $\rho$ and $\phi$ are implemented by MLPs, the same guarantee holds.*

If the encoder $S \mapsto \sum_{x \in S} \phi(x)$ is not injective over $\mathcal{X}^{\leq M}$, distinct sets may collapse to the same code, preventing any $\rho$ from separating them. Thus, universal approximation in practice requires sufficiently expressive $\phi$ (dimension $p$ and nonlinearity) so that the induced embedding is injective on the relevant family. By classical approximation results (e.g., Stone–Weierstrass), MLP families are dense in $C(K)$ on compact $K$, yielding the stated universality.

## C  PROOF OF THEOREM 5.2

**Theorem C.1** *Given an ILP instances $\mathcal{I}$, $\mathcal{G}_n = (V, E)$ is $\mathcal{I}$'s variable graph and $m$ denotes the number of constrains in $\mathcal{I}$, if the $i-th$ bag $X_i$ of vertex subsets $X = \{X_1, X_2, \cdots X_k\}(X_i \subseteq V)$ is formed by all the variable nodes involved in the $i-th$ corresponding constraint in $\mathcal{I}$, then it yields a graph decomposition of a graph $\mathcal{G}_n = (V, E)$.*

Recall the definition a graph decomposition, we will prove that regard the constrain node as the bag of variable yields a graph decomposition. We verify the three requirements:

*Vertex coverage.* We first assume that every variable node appears in at least one constraint, then by assumption, $v \in V$ appears in at least one constraint, hence $v_i \in X_j$ for some $j \in [m]$. Therefore $\bigcup_{j=1}^{m} X_r = V$. If there is variable $x_i$ does not appear in any constraint, we can handle it in the following equivalent ways: **Eliminate the variable from the ILP:** Optimize $x_i$ independently according to the objective coefficient $c_i$ (set $x_i^\star = 0$ if $c_i < 0$, $x_i^\star = 1$ if $c_i \geq 0$), fix $x_i = x_i^\star$, and remove $x_i$ from the ILP. This yields an equivalent ILP that no longer contains $x_i$, after which the preceding vertex-coverage argument applies unchanged.

*Edge coverage.* Take any edge $(v_{i_1}, v_{i_2}) \in E$. By the definition of the variable graph, there exists a constraint $j$ in which $v_{i_1}$ and $v_{i_2}$ co-occur. By construction, both $v_{i_1}$ and $v_{i_2}$ belong to the bag $X_j$, so $\{u, v\} \subseteq X_r$.

*Structural relation.* By definition of $\Xi$, two bags are adjacent exactly when they intersect, i.e., if $X_i \cap X_j \neq \emptyset$ then $(i, j) \in \Xi$. This matches the stated structural condition.

All three conditions in the definition are satisfied, so $(X, \Xi)$ is a graph decomposition of $\mathcal{G}_n$.

## D  PROOF OF THEOREM 5.3

**Theorem D.1** *Given any ILP instance $\mathcal{I}$, $\mathbf{x}^t \in \{0,1\}^n$ is the incumbent solution, $h^t \in \{0,1\}^n$ as neighborhood selection indicator generated from first stage and $\mathcal{M}(\mathcal{I}, \mathbf{x}, h^t)$ as the neighbors searching space, then $\forall \epsilon, \phi > 0$, there is $F \in \mathcal{F}_{CE_GNN}$ such that the following holds:*

- *For effective and redundant indicator $\mathcal{R}(\mathcal{I}, \mathbf{x}^t, h), \mathcal{T}(\mathcal{I}, \mathbf{x}^t, h)$, we have*

$$P(\|\hat{\mathcal{R}} - \mathcal{R}(\mathcal{I}, \mathbf{x}^t, h)\| > \phi) < \epsilon, \quad P(\|\hat{\mathcal{T}} - \mathcal{T}(\mathcal{I}, \mathbf{x}^t, h)\| > \phi) < \epsilon$$

  *where $\hat{\mathcal{R}}, \hat{\mathcal{T}}$ are probabilistic predictions for effective and redundant indicator generated from the second stage.*

- *Denote $\tilde{h}^t \in [0,1]^n$ as the output from $F$ after calibration, if the scale of neighborhood $k^t < \frac{n}{2}$, then we have*

$$P(\|\tilde{h}^t - h_0^t\| > \phi) < \epsilon$$

*where $h_0^t$ is optimal neighborhood selection indicator:$\max_{h \in LB(k^t)} \left( E(\mathcal{I}, \mathbf{x}^t, h) \right)$ ($E(\mathcal{I}, \mathbf{x}^t, h), LB(k)$ is defined in equation ??)*

## D.1 PROOF OF PART I OF THE THEOREM

In this section, we prove the results in Section , i.e., our framework is able to approximate the predictions of effectiveness of constraint and optimal neighbor on finite datasets of ILPs with arbitrarily small error. We consider more general results on finite-measure subset of which involves the infinite elements. In our settings, the predictions of effectiveness of constraint only depends on the ILP instance $\mathcal{I}$, current solution $x^0$ and prediction of optimal neighbor $h \in \{0,1\}^n$, while the optimal neighbor only depends on the ILP instance $\mathcal{I}$ and current solution $x^0$. Therefore, we are able to define the mapping for predictions of local-effectiveness of constraint as follow:

$$\Phi_{loc-ex} : \mathbb{R}^{n \times m} \times \mathbb{R}^m \times \mathbb{R}^n \times \{0,1\}^n \times \{0,1\}^n \to \{0,1\}^n : \mathcal{I} \times x^0 \times h \to r \quad (7)$$

and the mapping for optimal neighbor as follow:

$$\Phi_{opti} : \mathbb{R}^{n \times m} \times \mathbb{R}^m \times \mathbb{R}^n \times \{0,1\}^n \to \{0,1\}^n : \mathcal{I} \times x^0 \to \{0,1\}^n \quad (8)$$

Where $\mathcal{I}$ refers to $(A,b,c) \in \mathbb{R}^{n \times m} \times \mathbb{R}^m \times \mathbb{R}^n$, $x^0 \in \{0,1\}^n$. Without loss of generality, the effectiveness of a constraint is permutation-invariant with respect to the ordering of the variables. Reorder the variables such that the first $n_1$ variables are relaxed and the remaining $n_1 = n - n_2$ variables are fixed. Denote $A_{h,1} \in \mathbb{R}^{n_1 \times m}$ as the induced submatrix by selecting the columns of $A$ whose corresponding entry in $h$ equals 1, and $A_{h,0} \in \mathbb{R}^{n_1 \times m}$ as equals 1. Also denote $c_{h,1} \in \mathbb{R}^{n_1}$ as the induced subvector by selecting the elements of $c$ whose corresponding entry in $h$ equals 1. Thus, the original problem is reformulated as a reduced-scale ILP problem:

$$\min_x \quad c_{h,1}^\top x \quad (9)$$

$$\text{s.t.} \quad A_{h,1} x \leq b - A_{h,0} x_{h,0}^0, \quad (10)$$

$$x \in \{0,1\}^{n_1}. \quad (11)$$

Under the relaxation condition specified by $h$, the ILP subproblem(denoted as $\mathcal{I}_{x^0,h}$) corresponding to ILP problem $\mathcal{I}$ is unique, hence we are able to define a measurable mapping $\Phi_{sub}$:

$$\Phi_{indu\_cur}(\mathcal{I}, x^0, h) = \mathcal{I}_{x^0,h} : \mathbb{R}^{n \times m} \times \mathbb{R}^m \times \mathbb{R}^n \times \mathbb{R}^n \times \mathbb{R}^n \to \mathbb{R}^{n_1 \times m} \times \mathbb{R}^m \times \mathbb{R}^{n_1} \quad (12)$$

To prove $\Phi_{ex}$ and $\Phi_{opti}$ is measurable, we first define the mapping of optimal-feasibility $\Phi_{\text{feas}}$:

$$\Phi_{opti\_feas}(\mathcal{I}, x^0) : \mathbb{R}^{n \times m} \times \mathbb{R}^m \times \mathbb{R}^n \to \{0,1\} \quad (13)$$

That equals to 1 if it has a better feasible solution than $x^0$ and 0 otherwise, then we have the following lemma.

### D.1.1 PROOF OF LEMMA $D.2$ AND $D.3$

**Lemma D.2** *The optimal-feasibility mapping $\Phi_{feas}$ defined in 13 is measurable, i.e., the preimages $\Phi_{opti\_feas}^{-1}(1)$ and $\Phi_{opti\_feas}^{-1}(0)$ are both measurable subsets of $\mathcal{I}$.*

Proof of lemma $D.2$: Since $\Phi_{opti\_feas}^{-1}(1) \cup \Phi_{opti\_feas}^{-1}(0)$ refers to every ILP, hence we only need to prove that $\Phi_{opti\_feas}^{-1}(1)$ is measurable. Hence we define the following measurable set:

$$X_{opti\_feas} = \{\mathcal{I}, x^0 \in \mathbb{R}^{n \times m} \times \mathbb{R}^m \times \mathbb{R}^n \times \mathbb{R}^n : \exists x \in \{0,1\}^n, s.t. Ax \leq b, c^\top x < c^\top x^0\} \quad (14)$$

Therefore the feasibility mapping $\Phi_{opti\_feas}$ defined in 13 is measurable.

Now we can prove that the mapping $\Phi_{loc-ex}$ is measurable: For $j \in [m]$, consider the following ILP instance, which is denoted as $\mathcal{I}_j^{x^0}$:

$$\min_x \quad c_{h,1}^\top x \tag{15}$$

$$\text{s.t.} \quad A_{h,1}^j x \leq b_j - A_{h,0}^j x_{h,0}^0, \tag{16}$$

$$x \in \{0,1\}^{n_1}. \tag{17}$$

where $A_{h,1}^j$ and $A_{h,0}^j$ refers to the $j^{th}$ constrain. By lemma $D.2$, the $\Phi_{\text{feas}}(\mathcal{I}_j^{x^0})$ is measurable. In other way, since the $\mathcal{I}_j^{x^0}$ is induced from ILP $\mathcal{I}^{x^0}$

$$\min_x \quad c_{h,1}^\top x \tag{18}$$

$$\text{s.t.} \quad A_{h,1} x \leq b - A_{h,0} x_{h,0}^0, \tag{19}$$

$$x \in \{0,1\}^{n_1}. \tag{20}$$

therefore the mapping

$$\Phi_{indu\_con}(\mathcal{I}^{x^0}, j) = \mathcal{I}_j^{x^0} : \mathbb{R}^{n_1 \times m} \times \mathbb{R}^m \times \mathbb{R}^{n_1} \to \mathbb{R}^{n_1} \times \mathbb{R} \times \mathbb{R}^{n_1} \tag{21}$$

is measurable. In summary, denote

$$\Phi_{tran}(\mathcal{I}, x^0, h, j) = \Phi_{opti\_feas}(\Phi_{indu\_con}(\Phi_{indu\_cur}(\mathcal{I}, x^0, h), j)) \tag{22}$$

the mapping

$$\Phi_{tran}(\mathcal{I}, x^0, h, 1) \oplus \Phi_{tran\_}(\mathcal{I}, x^0, h, 2) \oplus \cdots \oplus \Phi_{tran\_}(\mathcal{I}, x^0, h, m) \tag{23}$$

is same as $\Phi_{loc-ex}$ therefore we have proved the following lemma:

**Lemma D.3** *The local-effectiveness of constraint mapping $\Phi_{loc-ex}$ defined in 7 is measurable.*

### D.1.2 STONE-WEIERSTRASS THEOREM

Stone-Weierstrass theorem describes that any continuous function defined on a closed interval can be uniformly approximated by polynomial functions. Its formal statement is as follow:

**Theorem D.4 (Stone-Weierstrass Theorem)** *Let $X$ be a compact Hausdorff space and let $\mathcal{A} \subseteq C(X, \mathbb{R})$ be a subalgebra of the algebra of continuous real-valued functions on $X$. Suppose that:*

1. *$\mathcal{A}$ separates points, i.e., for any $x, y \in X$ with $x \neq y$, there exists $f \in \mathcal{A}$ such that $f(x) \neq f(y)$;*

2. *$\mathcal{A}$ contains the constant functions.*

*Then $\mathcal{A}$ is dense in $C(X, \mathbb{R})$ with respect to the uniform norm. In other words, for every $f \in C(X, \mathbb{R})$ and every $\varepsilon > 0$, there exists $g \in \mathcal{A}$ such that*

$$\sup_{x \in X} |f(x) - g(x)| < \varepsilon.$$

To establishing universal approximation for $\Phi_{loc-ex}$, there are two main lemma remaining to be proved.

- $\Phi_{loc-ex}$ is a graph-invariant function.

- The Weisfeiler–Lehman (WL) test possesses sufficient discriminative capability to recognize local-effective constraints.

### D.1.3 PROOF OF LEMMA $D.5$

**Lemma D.5** *Let $\chi(b_{j_1})$ and $\chi(b_{j_2})$ be the output label of constraint nodes $b_1$ and $b_2$ in the WL test after adding the features of variable relaxation/fixation, if $\chi(b_{j_1}) = \chi(b_{j_2})$ then their local effectivity is consistent.*

Proof of lemma $D.5$:Recall of the process of WL-test, since $\chi(b_{j_1}) = \chi(b_{j_2})$, then we have

$$(\chi(b_{j_1}), \{\{(\chi(x_i), e_{i,j_1}) | x_i \in N(b_{j_1})\}\}) = (\chi(b_{j_2}), \{\{(\chi(x_i), e_{i,j_2}) | x_i \in N(b_{j_2})\}\}) \qquad (24)$$

where $N(b_j)$ denotes the neighbor of node $b_j$, since the WL test take the features of variable relaxation/fixation into consideration, therefore $N(b_j)$ can be partitioned into two parts based on the features of variables relaxation/fixation: $N_{h=1}(b_j)$ refers to the variables that is relaxed while $N_{h=0}(b_j)$ refers to the variables that is fixed. Also we have

$$\forall x_{i_1} \in N_{p=1}(b_j), x_{i_2} \in N_{h=0}(b_j), \; \chi(x_{i_1}) \neq \chi(x_{i_2}) \qquad (25)$$

Take equation 25 into equation 24, then we have

$$\{\{(\chi(x_i), e_{i,j_1}) | x_i \in N_{h=0}(b_{j_1})\}\} = \{\{(\chi(x_i), e_{i,j_2}) | x_i \in N_{h=0}(b_{j_2})\}\} \qquad (26)$$

According to the WL process, $\chi(x_{i_1}) = \chi(x_{i_2})(\chi(x_{i_1}) \in N_{h=0}(b_{j_1}), x_{i_2} \in N_{h=0}(b_{j_2}))$ suggests the equivalence between the input features of variables $x_{i_1}$ and $x_{i_2}$: (1) the current solution value of $x_{i_1}$ and $x_{i_2}$ (2) the coefficient of $(x_{i_1}, b_{j_1}) : a_{i_1,j_1}$ and $(x_{i_2}, b_{j_2}) : a_{i_2,j_2}$ in the constraint matrix $A$. Therefore, $A_{h,0}^{j_1} x_{h,0}^0 = A_{h,0}^{j_2} x_{h,0}^0$.

Suppose constrain $b_{j_1}$ is local-effective then ILP $\mathcal{I}_{j_1}^{x^0}$:

$$\min_x \quad c_{h,1}^\top x \qquad (27)$$
$$\text{s.t.} \quad A_{h,1}^{j_1} x \leq b_{j_1} - A_{h,0}^{j_1} x_{h,0}^0, \qquad (28)$$
$$x \in \{0,1\}^{n_1}. \qquad (29)$$

has a better solution, denoted as $x^{j_1}$, than current solution. Note that

$$\{\{(\chi(x_i), e_{i,j_1}) | x_i \in N_{h=1}(b_{j_1})\}\} = \{\{(\chi(x_i), e_{i,j_2}) | x_i \in N_{h=1}(b_{j_2})\}\} \qquad (30)$$

therefore for ILP $\mathcal{I}_{j_2}^{x^0}$:

$$\min_x \quad c_{h,1}^\top x \qquad (31)$$
$$\text{s.t.} \quad A_{h,1}^{j_2} x \leq b_{j_2} - A_{h,0}^{j_2} x_{h,0}^0, \qquad (32)$$
$$x \in \{0,1\}^{n_1}. \qquad (33)$$

the better solution is assigned based on the following strategy:

- Rearrange the variables in $N_{h=1}(b_{j_1})$ and $N_{h=1}(b_{j_2})$ as $(x_1^{j_1}, x_2^{j_1}, \cdots)$ and $(x_1^{j_2}, x_2^{j_2}, \cdots)$ that $\chi(x_i^{j_1}) = \chi(x_i^{j_2})$. The implementation is feasible according to Equation 30.

- Assign the value of solution $x^{j_2}$ to $x_i^{j_2}$ same as $x_i^{j_1}$.

Similar to $N_{h=0}$, $\chi(x_{i_1}) = \chi(x_{i_2})(\chi(x_{i_1}) \in N_{h=1}(b_{j_1}), x_{i_2}) \in N_{h=1}(b_{j_2}))$ suggests the equivalence between the input features of variables $x_{i_1}$ and $x_{i_2}$: (1) the coefficient of $(x_{i_1}, b_{j_1}) : a_{i_1,j_1}$ and $(x_{i_2}, b_{j_2}) : a_{i_2,j_2}$ in the constraint matrix $A$. Therefore, $A_{h,0}^{j_1} x_{h,0}^0 = A_{h,0}^{j_2} x_{h,0}^0$. (2) objective coefficient vector: $c_{i_1} = c_{i_2}$. Due to (1) $A_{h,1}^{j_1} x^{j_1} = A_{h,1}^{j_2} x^{j_2}$, therefore $x^{j_2}$ is feasible for ILP $\mathcal{I}_{j_2}^{x^0}$. Due to (2) $c^\top x^{j_2} = c^\top x^{j_1} < c^\top x^0$, $x^{j_2}$ is a better solution, hence constrain $b_{j_2}$ is also local-effective. Now we have proved that if $\chi(b_{j_1}) = \chi(b_{j_2})$ then their local effectivity is consistent. Now we are proving the other lemma for establishing universal approximation for $\Phi_{loc-ex}$.

### D.1.4 PROOF OF LEMMA $D.6$

**Lemma D.6** $\Phi_{loc-ex}$ *is a graph-invariant function with respect to variable permutation and graph-equivariant function with respect to constrain permutation. In other word, denote $\pi(\mathcal{I}), \pi(x^0), \pi(h)$ as $\mathcal{I}, x^0, h$ after applying permutation $\pi$, if permutation $\pi$ acts on variable then $\Phi_{loc-ex}(\pi(\mathcal{I}), \pi(x^0), \pi(h)) = \Phi_{loc-ex}(\mathcal{I}, x^0, h)$, if permutation $\pi$ acts on constrain then $\Phi_{loc-ex}(\pi(\mathcal{I}), \pi(x^0), \pi(h)) = \pi(\Phi_{loc-ex}(\mathcal{I}, x^0, h))$.*

Proof of lemma $D.6$: Given an ILP instance $\mathcal{I}$:

$$\min_x \quad c^\top x \tag{34}$$

$$\text{s.t.} \quad Ax \leq b, \tag{35}$$

$$x \in \{0,1\}^n. \tag{36}$$

For bipartite graphs, we will show the invariance with respect to variable nodes and the equivariance with respect to constraint nodes for $\Phi_{loc-ex}$. We first discuss the invariance with respect to variable nodes.

For $\pi_n \in S_n$, let $P_\pi \in \{0,1\}^{n \times n}$ be its permutation matrix, then under the action of the permutation that reorders the variable nodes, the new coefficient adjacency matrix is $AP_\pi$, the new constraint coefficient vector is $b$ and the new objective coefficient vector is $c^\top P_\pi$, and the corresponding current solution $P_\pi^\top x^0$. If the constraint $b_j$ before permutation is effective in the neighborhood determined by $h$, then it has a better solution, denote $x_j^h$ as it associated with the variables being fixed, then $P_\pi^\top x_j^p$ is also a better solution for the ILP after variable permutation.

As for the equivariance with respect to constraint nodes, for $\pi_m \in S_m$, let $P_\pi \in \{0,1\}^{n \times n}$ be its permutation matrix, if constraint $b_j$ has a better solution, denote $x_j^p$ then $x_{\pi(j)}^h$ is a better solution for constraint $b_{\pi(j)}$, therefore $\Phi_{loc-ex}(\pi(\mathcal{I}, x^0, p)) = \pi(\Phi_{loc-ex}(\mathcal{I}, x^0, p))$. Now we have proved the lemma $D.6$.

In summary, we now can prove there is a model $\mathcal{F} \in F_C$ can universally approximate $\Phi_{loc-ex}$: By lemma $D.3$ $\Phi_{loc-ex}$ is measurable, while by lemma $D.5$ we have that if the label of two constrain in WL test are same then their local effectiveness is consistent and the invariance/equivariance by $D.6$. Therefore by Stone-Weierstrass theorem there is a model $\mathcal{F} \in F_C$ can universally approximate $\Phi_{loc-ex}$.

## D.2 PROOF OF PART II OF THE THEOREM

(2)The proof of the second part of the theorem can be summarized in the following three points.

- First, similar to foldable-ILP, we given the definition for foldable-ILP, and prove that for the unfoldable case, the WL test will eventually produce a unique discrete coloring, and therefore if two graphs cannot be distinguished by the WL test, they must be isomorphic.

- By applying Lusin's theorem: any measurable function on a set of finite measure can be approximated by a continuous function on almost all points. By using the Stone–Weierstrass type theorem: on a compact set, GNNs can uniformly approximate all continuous mappings whose separation power does not exceed that of the WL test.

- In our framework, the mechanism that makes neighborhood selection decisions based on the GNN output probabilities in the third stage further differentiates the nodes with identical labels in the WL test.

Similar to foldable-ILP, we give the definition of foldable-ILP:

**Definition D.7 (Foldable ILP)** *Given any ILP instance $\mathcal{I}$, we say that $\mathcal{I}$ is* foldable *if, by running the WL test on its corresponding bipartite graph, there exist two variate nodes, their labels are for any choice of hash functions in the WL test. The unfoldable ILP is the rest ILP that is not foldable.*

*The collection of foldable ILP instances are denoted as $\mathbb{I}_{fold} \subset G_{m,n} \times H_m^V \times H_n^W$.*

To establish that our framework is capable of approximating the output optimal neighborhood. We need to prove that the separation power of the WL test is stronger than that of the function $\Phi_{opti}$ for unfoldable ILP instances. Therefore we have the following lemma:

### D.2.1 PROOF OF LEMMA $D.8$

**Lemma D.8** *For any two unfoldable ILP instances with current solution as variable nodes' extra feature, then if their corresponding bipartite graphs $(G_1, G_2)$ are isomorphic if and only if the WL test determines that $(G_1, G_2)$ are isomorphic.*

Proof of lemma $D.8$: It's trivial that the isomorphism of $(G_1, G_2)$ implies the isomorphism in WL test.

To prove the isomorphism of $(G_1, G_2)$ in WL test implies the isomorphism. Since $G_1, G_2$ are unfoldable, then any label of nodes in $G_1, G_2$ is unique and WL test determines that $(G_1, G_2)$ are isomorphic, it yeild a bijection mapping $f : V_1 \to V_2$, that $\forall v^1 \in V_1, \chi(v^1) = \chi(f(v^1))$. We will prove that $f$ is a isomorphic mapping, as for every pair of nodes $(v_{i_1}^1, v_{i_2}^1)$ in $G_1$, if $(v_{i_1}^1, v_{i_2}^1) \in E_1$, then $(f(v_{i_1}^1), f(v_{i_2}^1)) \in E_2$:

Since $(G_1, G_2)$ are isomorphic in WL test. By WL test condition, there's a corresponding pair of node $(v_{i_1}^2, v_{i_2}^2)$ in $G_2$ that $\chi(v_{i_1}^1) = \chi(v_{i_1}^2)$ and $\chi(v_{i_2}^1) = \chi(v_{i_2}^2)$. Suppose $(v_{i_1}^2, v_{i_2}^2) \notin E_2$, then since $\chi(v_{i_1}^1) = \chi(v_{i_1}^2)$, recall of the process of WL test:

$$(\chi(v_{i_1}^1), \{\{(\chi(u^1), e_{(v_{i_1}^1, u^1)}| u^1 \in N(v_{i_1}^1)\}\}) = (\chi(v_{i_1}^2), \{\{(\chi(u^2), e_{(v_{i_1}^2, u^2)}| u^2 \in N(v_{i_1}^2)\}\}) \quad (37)$$

Since $G_1, G_2$ are unfoldable, $\chi(v_{i_2}^1)$ and $\chi(v_{i_2}^2)$ are unique. On other hand, $\chi(v_{i_2}^1) \in N(v_{i_1}^1)$, therefore if $(v_{i_1}^2, v_{i_2}^2) \notin E_2$, then

$$\{\{(\chi(u^1), e_{(v_{i_1}^1, u^1)}| u^1 \in N(v_{i_1}^1)\}\} \neq \{\{(\chi(u^2), e_{(v_{i_1}^2, u^2)}| u^2 \in N(v_{i_1}^2)\}\} \quad (38)$$

therefore $(v_{i_1}^2, v_{i_2}^2) \notin E_2$. Hence we have $(v_{i_1}^1, v_{i_2}^1) \in E_1$ if and only if $(f(v_{i_1}^1), f(v_{i_2}^1)) \in E_2$, which proves lemma $D.8$.

### D.2.2 PROOF OF MEASURABILITY FOR OPTIMAL NEIGHBOR MAPPING

**Optimal Neighbor Mapping.** For any unfoldable ILP instance $\mathcal{I}$ with current solution $x^0$, the associated ILP problem has a finite optimal objective value. Although an ILP may admit multiple optimal neighbors, it is guaranteed that there exists a unique optimal neighbor with the smallest $\ell_2$-norm. Formally, we define the mapping

$$\Phi_{neigh}(\mathcal{I}, x^0) = p : (\mathbb{R}^{n \times m} \times \mathbb{R}^n \times \mathbb{R}^m \backslash \mathbb{I}_{fold}) \times \{0, 1\}^n \to \{0, 1\}^n,$$

which maps $(\mathcal{I}, x^0)$ to the $p$ indicator vector that decide the variable is fixed or relaxed with the smallest $\ell_2$-norm. $\Phi_{neigh}$ maps an ILP with current solution to exactly one of its optimal neighbor and we choose the $h$ the smallest $\ell_2$-norm as unique, otherwise ILP instance $\mathcal{I}$ is not unfoldable.

**Lemma D.9** *The optimal neighbor mapping $\Phi_{neigh}(\mathcal{I}, x^0)$ is measurable.*

Proof of lemma $D.9$: In Chen et al. (2023), chen has proved that the optimal solution and value mapping for mixed-integer linear programs(ILP) is measurable, since ILP is the subset of ILP therefore the optimal solution mapping for integer linear programs for unfoldable ILP

$$\Phi_{opti\_solu}(\mathcal{I}) : (\mathbb{R}^{n \times m} \times \mathbb{R}^n \times \mathbb{R}^m \backslash \mathbb{I}_{fold}) \to \{0, 1\}^n,$$

and value

$$\Phi_{opti\_value}(\mathcal{I}) : (\mathbb{R}^{n \times m} \times \mathbb{R}^n \times \mathbb{R}^m \backslash \mathbb{I}_{fold}) \to \mathbb{R},$$

are measurable. Since the mapping $\Phi_{indu\_cur}$ is also measurable, we can define the mapping $\Phi_{indu\_cur\_opti\_solu}$ that output the optimal solution and $\Phi_{indu\_cur\_opti\_value}$ that output the optimal value for the sub-ILP induced by $p$

$$\Phi_{indu\_cur\_opti\_solu}(\mathcal{I}, x^0, p) = \Phi_{opti\_solu}(\Phi_{indu\_cur\_opti\_solu}(\mathcal{I}, x^0, p)) \quad (39)$$

and

$$\Phi_{indu\_cur\_opti\_value}(\mathcal{I}, x^0, p) = \Phi_{opti\_value}(\Phi_{indu\_cur\_opti\_solu}(\mathcal{I}, x^0, p)) \tag{40}$$

Since the composition of measurable functions is also measurable, therefore $\Phi_{indu\_cur\_opti\_solu}$ and $\Phi_{indu\_cur\_opti\_value}$ are also measurable. Denote $c$ as

$$c = \max_{p}(\Phi_{indu\_cur\_opti\_value}(\mathcal{I}, x^0, p)) \tag{41}$$

then

$$\Phi_{indu\_cur\_opti\_value}^{-1}(c)_{\{\mathcal{I}, x^0\}} \tag{42}$$

outputs indicator $p$ of optimal neighbor with the smallest $\ell_2$-norm. Hence The optimal neighbor mapping $\Phi_{neigh}(\mathcal{I}, x^0)$ is measurable. To enable GNN to approximate the function, Chen et al. (2023) has shown the measurability for invariant and equivariant mapping, the theorem is as follow:

**Theorem D.10 (Theorem A.10 in Chen et al. (2023))** *Let $X \subset \mathbb{R}^{n \times m} \times \mathcal{H}^n \times \mathcal{H}^m$ be a compact subset that is closed under the action of $S_m \times S_n$. Suppose that $\Phi \in C(X, \mathbb{R}^n)$ satisfies:*

- *For any $\sigma_V \in S_m$, $\sigma_W \in S_n$, and $G \in X$,*

$$\Phi\big((\sigma_V, \sigma_W)G\big) = \sigma_W\big(\Phi(G)\big).$$

- *$\Phi(G) = \Phi(\hat{G})$ for all $G, \hat{G} \in X$ with*

$$G \overset{WL}{\sim} \hat{G}.$$

- *Given any $G \in X$ and any $i, i' \in \{1, 2, \ldots, n\}$, if $\chi(v_i) = \chi(v_{i'})$ holds for any choices of hash functions (i.e., the WL colors of node $v_i$ and $v_{i'}$ coincide at every iteration), then*

$$\Phi(G)_i = \Phi(G)_{i'}.$$

*Then for any $\varepsilon > 0$, there exists $F_W \in \mathcal{F}_{GNN}^W$ such that*

$$\sup_{(G) \in X} \big\|\Phi(G) - F_W(G)\big\| < \varepsilon.$$

Since the current solution $x^0$ can regarded as a extra feature for $\mathcal{I}$ in $\mathcal{H}^n$, therefore theorem $D.10$ can also be applied for $\Phi_{neigh}$ when $X$ is within the unfoldable ILP instance. By lemmas $D.8$, $D.9$ and theorem $D.10$, we have proved that for any unfoldable ILP instances can be approximated by MP-GNN.

### D.3 FOLDABLE ILP INSTANCES APPROXIMATED BY CE-FRAMEWORK

Now we can prove that our framework is also suitable for foldable ILP instances: Consider a foldable ILP $\mathcal{I}$, there are two following conditions:

- the foldable ILP only refers to constrains nodes. then there are no variable node $v_{i_1}$ and $v_{i_2}$ that $\chi(v_{i_1}) = \chi(v_{i_2})$.
- there are two variable nodes $v_{i_1}$ and $v_{i_2}$ that $\chi(v_{i_1}) = \chi(v_{i_2})$.

*Condition 1:* By definition of 1-WL on a weighted bipartite graph, at the stable coloring we have, for each constraint $u_j$,

$$\chi(u_j) = \text{Hash}\Big(\chi(u_j), \{\{(\chi(x_i), A_{i,j}) : x_i \in N(u_j)\}\}\Big).$$

Because $\chi$ is injective on $V_x$, the multiset $\{\{(\chi(x_i), A_{i,j})\}\}$ can be canonically reindexed by (unique) variable colors. Thus the WL signature of $u_j$ is exactly the *full incident coefficient profile to individually identified variables*. If two constraints $u_{j_1}, u_{j_2}$ satisfy $\chi(u_{j_1}) = \chi(u_{j_2})$, then their profiles coincide entrywise across all variables, i.e.,

$$A_{i,j_1} = A_{i,j_2} \quad \text{for every } x_i \in V_x.$$

Consequently, swapping $u_{j_1}$ and $u_{j_2}$ is an automorphism that fixes all variable nodes pointwise and preserves all edge weights. Any graph function $\Phi(G) \in \mathbb{R}^n$ that is invariant to constraint relabeling (and equivariant to constraint permutations), as is the case for $\Phi_{\text{neigh}}$, is therefore insensitive to such swaps.

Now consider two ILP with incumbent solution instances graphs $G, \hat{G}$ that are WL-equivalent with the same variable-injective stable coloring. The color-preserving correspondence fixes variables one-to-one. On the constraint side, it may include permutations among color-tied constraints, but as argued-those permutations do not affect any constraint–equivariant variable-output $\Phi$. Hence WL-equivalence under variable-injective coloring implies equivalence for $\Phi_{\text{neigh}}$'s input-output behavior. For variables-unfoldable instance, if $\chi(u_{j_1}) = \chi(u_{j_2})$ then the $j_1^{th}$ and $j_2^{th}$ serve have exactly the same constraint power over the variables. Specially, denote the original ILP instance as

$$min \quad \mathbf{c}^\top \mathbf{x} \tag{43}$$

$$s.t. \sum_{i \in [n]} A_{i,j} x_i \leq b_j, \quad j \in [m], j \neq j_1, j_2 \tag{44}$$

$$\sum_{i \in [n]} A_{i,j_1} x_i \leq b_{j_1} \tag{45}$$

$$\sum_{i \in [n]} A_{i,j_2} x_i \leq b_{j_2} \tag{46}$$

$$x_i \in \{0, 1\} \tag{47}$$

then ILP instance

$$min \quad \mathbf{c}^\top \mathbf{x} \tag{48}$$

$$s.t. \sum_{i \in [n]} A_{i,j} x_i \leq b_j, \quad j \in [m], j \neq j_1, j_2 \tag{49}$$

$$\sum_{i \in [n]} A_{i,j_1} x_i \leq b_{j_1} \tag{50}$$

$$x_i \in \{0, 1\} \tag{51}$$

has same solution as original ILP, while in this way the constrain which shares the same label in WL test can be cut off until the ILP is unfoldable.

Therefore, the only symmetry that could degrade separability relevant to $\Phi_{\text{neigh}}$ would be a tie on *variable* colors. Since $G$ is variable–unfoldable, no such tie exists, and the instance behaves (for our theory and approximation guarantees) exactly like an unfoldable ILP. If $G$ is variable-unfoldable, then any equality of WL colors on the constraint side (i.e., $\chi(u_j) = \chi(b_{j'})$) does not reduce the distinguishability of variables for any constraint-equivariant target $\Phi : \mathcal{G} \to \{0, 1\}^n$. In particular, $G$ can be treated as *unfoldable* for the purposes of approximating $\Phi_{\text{neigh}}$.

*Condition 2:* We first will prove that: *if foldable-on-variables(variable nodes share the same label in WL test) is under $p$-consistency*. In other word, if

$$\chi(v_{i_1}) = \chi(v_{i_2}) \rightarrow p_{i_1} = p_{i_2}.$$

Denote the label for WL test in the third stage as $\hat{\chi}$, since the extra added feature $p$ in WL test is consistent with label $\chi$, therefore we have

$$\chi(v_{i_1}) = \chi(v_{i_2}) \longleftrightarrow \hat{\chi}(v_{i_1}) = \hat{\chi}(v_{i_2})$$

and if $\hat{\chi}(v_{i_1}) = \hat{\chi}(v_{i_2})$. Denote the set of color of label $\hat{\chi}(v)$ get as $\mathcal{C}$. Since $\forall c_t \in \mathcal{C}$, if the neighborhood selection pick a node $v_{i_1}$ as $\hat{\chi}(v_{i_1}) = c_t$, then the neighborhood selection will pick every variable node $v$ that $\hat{\chi}(v) = c_t$. In this way the neighborhood selection degenerates from multiset

$$\{\{\chi(v) | v \in V\}\}$$

to set

$$\{c_t | c_t \in \mathcal{C}\}$$

unlike multiset, all elements in the set are unique, therefore the neighborhood selection is unique. Therefore if foldable-on-variables is under $p$-consistency then it can be treated as unfoldable ILP.

Now we consider the condition that *there are a pair of foldable-variables is not under $p$-consistency*: First, we assume the bipartite graph is a connected graph(else the ILP can be divided into r independent ILPs if the graph has r disconnected components). To prove the condition we have to prove that for (finite, simple) trees, the 1-dimensional Weisfeiler–Lehman (WL) color refinement test distinguishes non-isomorphic graphs; equivalently, two trees are WL-equivalent if and only if they are isomorphic. The theorem is as follow

### D.3.1 Proof of theorem $D.11$

**Theorem D.11** *Let $T_1, T_2$ be trees. Then the following are equivalent:*

- $T_1 \cong T_2$ *(graph isomorphism).*

- $T_1 \equiv_{\mathrm{WL}} T_2$ *(the stable 1-WL colorings agree up to color renaming).*

*In particular, 1-WL decides isomorphism on trees.*

The direction $T_1 \cong T_2 \Rightarrow T_1 \equiv_{\mathrm{WL}} T_2$ is immediate, since 1-WL is isomorphism-invariant.

For the converse, we show that on any tree, the stable 1-WL color of a vertex encodes exactly the isomorphism type of its rooted subtree. Root a tree $T$ at an arbitrary vertex $r$ and orient edges away from $r$. Define the *height* $\mathrm{ht}(v)$ of a vertex $v$ as the distance to the farthest descendant; leaves have height 0.

We prove by induction on $h = \mathrm{ht}(v)$ that after $h$ rounds of 1-WL refinement, the color of $v$ is a complete invariant of its rooted subtree $(T, v)$:

*Claim.* After round $h$, two vertices $v, w$ have the same color if and only if the rooted trees $(T, v)$ and $(T, w)$ are isomorphic.

*Base $h = 0$.* Leaves all have the same multiset of neighbor colors (just their parent, if any). Their color is determined uniformly, and the rooted subtree at a leaf is a single node. Thus color equality coincides with rooted-subtree isomorphism.

*Induction step.* Assume the claim holds for all heights $< h$. Let $v$ satisfy $\mathrm{ht}(v) = h$ with children $u_1, \ldots, u_k$ (all of height $< h$). In round $h$, the new color of $v$ is computed from its current color together with the multiset of the children's colors from round $h - 1$. By the induction hypothesis, each child's color at round $h - 1$ uniquely represents the rooted isomorphism type of its subtree $(T, u_i)$. Therefore, the multiset of child colors at round $h - 1$ encodes exactly the multiset of rooted subtree types attached to $v$. Since a rooted tree is determined (by the standard AHU decomposition) by the multiset of its children's rooted types, the new color of $v$ at round $h$ uniquely encodes the rooted isomorphism type of $(T, v)$. Conversely, if two rooted subtrees are isomorphic, they induce the same multiset of children's types and hence the same color. This proves the claim at height $h$.

Thus, after $H := \max_v \mathrm{ht}(v)$ rounds (at most the radius/diameter of $T$), the stable color of each vertex identifies the isomorphism type of its rooted subtree. In particular, the (multi)set of stable colors of neighbors of any vertex encodes the branch structure around that vertex. To compare two *unrooted* trees $T_1, T_2$, pick any vertex $r_1 \in T_1$ and $r_2 \in T_2$. If $T_1 \equiv_{\mathrm{WL}} T_2$, then there is a bijection between the stable colors in $T_1$ and $T_2$ preserving adjacency color multisets. Choosing $r_1$ and $r_2$ with the same stable color and proceeding level by level, the above characterization yields an isomorphism between the rooted trees $(T_1, r_1)$ and $(T_2, r_2)$, hence an unrooted graph isomorphism $T_1 \cong T_2$. Therefore, WL-equivalence implies isomorphism on trees. Now we have proved theorem $D.11$.

### D.3.2 Separating WL Test Labels via Additional Features

Denote the set of color of label $\hat{\chi}(v)$ get as $\hat{\mathcal{C}}$ after adding indicator $h$ as features, and $\mathcal{C}$ as original. Therefore, we can divide the color of $\hat{\mathcal{C}}$ based on the value of $h$: $\hat{\mathcal{C}} = \mathcal{C}_0 \cup \mathcal{C}_1$ where $\mathcal{C}_0$ refers the color when $h = 0$, and $\mathcal{C}_0$ refers the color when $h = 1$. It is obvious that $\mathcal{C}_0 \cap \mathcal{C}_1 = \emptyset$. Since the

neighborhood size is less than half of the total number of nodes:$k^t < \frac{n}{2}$. Then we have

$$|\{v_i|h_i = 0\}| > |\{v_i|h_i = 1\}| \tag{52}$$

which is equivalent to

$$|\{v_i|\chi(v_i) \in \mathcal{C}_0\}| > |\{v_i|\chi(v_i) \in \mathcal{C}_1\}| \tag{53}$$

We divide the condition into two cases:(1) The foldable variables are under $h$-consistency when they are all in cycle. In this case, foldable variables are not under $p$-consistency iff they are in the tree. By theorem $D.11$ and $B.2$, trees are WL-equivalent if and only if they are isomorphic, therefore WL have enough expressive power to distinguish non-isomorphic graph.

(2)There are foldable variables in cycle that are not under $h$-consistency. Then divide the variable node as $V = \mathcal{CY} \cup \mathcal{TR}$, where $\mathcal{CY}$ denote the set of the variable nodes in cycle, $\mathcal{TR}$ denote the set of the variable nodes in tree. If

$$|\{v_i|\chi(v_i) \in \mathcal{C}_0, v_i \in \mathcal{TR}\}| > |\{v_i|\chi(v_i) \in \mathcal{C}_1, v_i \in \mathcal{TR}\}| \tag{54}$$

then it yields as cases (1). If

$$|\{v_i|\chi(v_i) \in \mathcal{C}_0, v_i \in \mathcal{TR}\}| \leq |\{v_i|\chi(v_i) \in \mathcal{C}_1, v_i \in \mathcal{TR}\}| \tag{55}$$

then since

$$|\{v_i|\chi(v_i) \in \mathcal{C}_0\}| > |\{v_i|\chi(v_i) \in \mathcal{C}_1\}| \tag{56}$$

we have

$$|\{v_i|\chi(v_i) \in \mathcal{C}_0, v_i \in \mathcal{CY}\}| > |\{v_i|\chi(v_i) \in \mathcal{C}_1, v_i \in \mathcal{CY}\}| \tag{57}$$

Hence at least one color $c \in \mathcal{C}$ that

$$|\{v_i|\chi(v_i) = c, p_i = 0, v_i \in \mathcal{CY}\}| > |\{v_i|\chi(v_i) = c, p_i = 1, v_i \in \mathcal{CY}\}| \tag{58}$$

Hence $\forall v_{i_1}, v_{i_2} \in \mathcal{CY}, \hat{\chi}(v_{i_1}) = \hat{\chi}(v_{i_2}) \iff i_1 = i_2$, and by theorem $D.11$ and $B.2$, trees are WL-equivalent if and only if they are isomorphic, therefore despite there might be node $v_{i_1}, v_{i_2} \in \mathcal{TR}$, $\hat{\chi}(v_{i_1}) = \hat{\chi}(v_{i_2})$ WL still have enough expressive power to distinguish non-isomorphic graph. In summary, the condition of theorem 5.3, the neighborhood size is less than half of the total number of nodes:$k^t < \frac{n}{2}$, breaks the symmetry of nodes sharing the same label under the WL test, thus enabling 1-WL to distinguish them. By lemmas $D.8$, $D.9$ and theorem $D.10$, any unfoldable ILP instances can be approximated by MP-GNN. Now that we the second part of theorem 5.3:Denote $\tilde{h}^t \in [0,1]^n$ as the output from $F$ after calibration, if the scale of neighborhood $k^t < \frac{n}{2}$, then we have

$$P(\|\tilde{h}^t - h_0^t\| > \phi) < \epsilon$$

where $h_0^t$ is optimal neighborhood selection indicator:$\max_{h \in LB(k^t)} \left( E(\mathcal{I}, \mathbf{x}^t, h) \right)$ $(E(\mathcal{I}, \mathbf{x}^t, h), LB(k)$ is defined in equation **??**)

# E  DETAILS OF NETWORK ARCHITECTURE

We give full details of the GAT and GCN architecture described in the following:

## E.1  GRAPH ATTENTION NETWORK

### E.1.1  STAGE I

The Network takes as input the state $s^t$ and output a score vector $F_\theta(s^t) \in [0,1]^n$, one score per variable. We use 2-layer MLPs with 64 hidden units per layer and ReLU as the activation function to map each node feature and edge feature to $\mathbb{R}^d$ where $d = 64$.

Let $\mathbb{X}_j, \mathbb{B}_i, \mathbb{E}_{i,j} \in \mathbb{R}^d$ be the embeddings of the $j$-th variable, $i$-th constraint and the edge connecting them output by the embedding layers. We perform two rounds of message passing through the GAT. In the first round, each constraint node $\mathbb{B}_i$ attends to its neighbors $N_i$ using an attention stucture with $H = 8$ attention heads:

$$\mathbb{B}_i' = \frac{1}{H} \sum_{h=1}^{H} \left( \alpha_{ii,1}^{(h)} \theta_{b,1}^{(h)} \mathbb{B}_i + \sum_{j \in N_i} \alpha_{ij,1}^{(h)} \theta_{x,1}^{(h)} \mathbb{X}_j \right) \tag{59}$$

where $\theta_{b,1}^{(h)} \in \mathbb{R}^{d \times d}$ and $\theta_{x,1}^{(h)} \in \mathbb{R}^{d \times d}$ are learnable weights. The updated constraint embeddings $\mathbb{B}_i$ are averaged across $H$ attention heads using attention weights Brody et al. (2021)

$$\alpha_{ij,1}^{(h)} = \frac{\exp(w_1^\mathsf{T} \rho([\theta_{b,1}^{(h)}\mathbb{B}_i, \theta_{x,1}^{(h)}\mathbb{X}_j, \theta_{e,1}^{(h)}\mathbb{E}_{i,j}]))}{\sum_{k \in N_i} \exp(w_1^\mathsf{T} \rho([\theta_{b,1}^{(h)}\mathbb{B}_i, \theta_{x,1}^{(h)}\mathbb{X}_k, \theta_{e,1}^{(h)}\mathbb{E}_{i,k}]))} \tag{60}$$

where the attention coefficients $w_1 \in \mathbb{R}^{3d}$ and $\theta_{e,1}^{(h)} \in \mathbb{R}^{d \times d}$ are both learnable weights and $\rho(\cdot)$ refers to the LeakyReLU activation function with negative slope 0.2. In the second round, similary, each variable node attends to its neighbors to get updated variable node embeddings

$$\mathbb{X}_j' = \frac{1}{H} \sum_{h=1}^{H} \left( \alpha_{jj,2}^{(h)} \theta_{x,2}^{(h)} \mathbb{X}_j + \sum_{i \in N_j} \alpha_{ji,2}^{(h)} \theta_{b,2}^{(h)} \mathbb{B}_i' \right) \tag{61}$$

with attention weights

$$\alpha_{ji,2}^{(h)} = \frac{\exp(w_2^\mathsf{T} \rho([\theta_{b,2}^{(h)}\mathbb{B}_i', \theta_{x,2}^{(h)}\mathbb{X}_j, \theta_{e,2}^{(h)}\mathbb{E}_{i,j}]))}{\sum_{k \in N_j} \exp(w_2^\mathsf{T} \rho([\theta_{b,2}^{(h)}\mathbb{B}_k', \theta_{x,2}^{(h)}\mathbb{X}_j, \theta_{e,2}^{(h)}\mathbb{E}_{k,j}]))} \tag{62}$$

where $w_2 \in \mathbb{R}^{3d}$ and $\theta_{b,2}^{(h)}, \theta_{v,2}^{(h)}, \theta_{e,2}^{(h)} \in \mathbb{R}^{d \times d}$ are learnable weights. After the two rounds of message passing, the final representations of variables $\mathbb{X}_j$ are passed through a 2-layer MLP with 64 hidden units per layer to obtain a scalar value for each variable. Finally, we apply the sigmoid function to get a score between 0 and 1.

### E.1.2 STAGE II

After yields a neighborhood selection indicator $h^t \in \{0,1\}^n$, the Network use 2-layer MLPs with 16 hidden units per layer and ReLU as the activation function to map $h^t$ to $\tilde{\mathbb{X}}_j \in \mathbb{R}^{d'}$ where $d' = 16$. Then network regard $\tilde{\mathbb{X}}_j$ as a extra feature and concatenate it with the original input: $[\mathbb{X}_j || \tilde{\mathbb{X}}_j]$. We perform a round of message passing through the fine-tuned GAT's first round in stage I as:

$$\mathbb{B}_i^{'II} = \frac{1}{H} \sum_{h=1}^{H} \left( \alpha_{ii,1}^{(h)} \theta_{b,1}^{(h)} \mathbb{B}_i + \sum_{j \in N_i} \alpha_{ij,1}^{(h)} [\theta_{x,1}^{(h)} || \tilde{\theta}_{x,1}^{(h)}][\mathbb{X}_j || \tilde{\mathbb{X}}_j] \right) \tag{63}$$

where weights $\theta_{b,1}^{(h)}$ and $\theta_{x,1}^{(h)}$ are fixed and $\tilde{\theta}_{x,1}^{(h)} \in \mathbb{R}^{d \times d'}$ learnable weights.

$$\alpha_{ij,1}^{(h)} = \frac{\exp([w_1, \tilde{w}_1]^\mathsf{T} \rho([\theta_{b,1}^{(h)}\mathbb{B}_i, \theta_{x,1}^{(h)}\mathbb{X}_j, \theta_{e,1}^{(h)}\mathbb{E}_{i,j}, \tilde{\theta}_{x,1}^{(h)}\tilde{\mathbb{X}}_j]))}{\sum_{k \in N_i} \exp([w_1, \tilde{w}_1]^\mathsf{T} \rho([\theta_{b,1}^{(h)}\mathbb{B}_i, \theta_{x,1}^{(h)}\mathbb{X}_k, \theta_{e,1}^{(h)}\mathbb{E}_{i,k} \tilde{\theta}_{x,1}^{(h)}\tilde{\mathbb{X}}_k]))} \tag{64}$$

where the attention coefficients $\tilde{w}_1 \in \mathbb{R}^{d'}$ and other weights are both fixed. The embedding $\mathbb{B}_i^{'II}$ goes a 2-layer MLPs with 64 hidden units per layer and ReLU as the activation function to obtain two scalar values for each constrain to predict the feauture of effective/redundant constraint. The newly introduced learnable parameters are updated by calibrating against the constraint features during training.

### E.1.3 STAGE III

Denote constrain feature as $r^t \in \{0,1\}^{2 \times m}$, the Network use 2-layer MLPs with 16 hidden units per layer and ReLU as the activation function to map $r^t$ to $\tilde{\mathbb{B}}_i \in \mathbb{R}^{d'}$ where $d' = 16$. Then network regard $\tilde{\mathbb{B}}_j$ as a extra feature and concatenate it with the original input: $[\mathbb{B}'_i || \tilde{\mathbb{B}}_i]$. We perform a round of message passing through the fine-tuned GAT's second round in stage I as:

$$\mathbb{X}_j^{III'} = \frac{1}{H} \sum_{h=1}^{H} \left( [\theta_{x,2}^{(h)} || \tilde{\theta}_{x,2}^{(h)}][\mathbb{X}_j || \tilde{\mathbb{X}}_j] + \sum_{i \in N_j} [\theta_{b,2}^{(h)} || \tilde{\theta}_{b,2}^{(h)}][\mathbb{B}'_i || \tilde{\mathbb{B}}_i] \right) \tag{65}$$

where weights $\theta_{b,2}^{(h)}$ and $\theta_{x,2}^{(h)}$ are fixed and $\tilde{\theta}_{x,2}^{(h)}, \tilde{\theta}_{b,2}^{(h)} \in \mathbb{R}^{d \times d'}$ learnable weights with attention weights:

$$\alpha_{ji,2}^{(h)} = \frac{\exp([w_2||\tilde{w}_2]^{\mathsf{T}} \rho([\theta_{b,2}^{(h)}\mathbb{B}'_i, \theta_{x,2}^{(h)}\mathbb{X}_j, \theta_{e,2}^{(h)}\mathbb{E}_{i,j}, \tilde{\theta}_{x,2}^{(h)}\tilde{\mathbb{X}}_j, \tilde{\theta}_{b,2}^{(h)}\tilde{\mathbb{B}}_i]))}{\sum_{k \in N_j} \exp(w_2^{\mathsf{T}} \rho([\theta_{b,2}^{(h)}\mathbb{B}'_k, \theta_{x,2}^{(h)}\mathbb{X}_j, \theta_{e,2}^{(h)}\mathbb{E}_{j,k}, \tilde{\theta}_{x,2}^{(h)}\tilde{\mathbb{X}}_j, \tilde{\theta}_{b,2}^{(h)}\tilde{\mathbb{B}}_k]))} \tag{66}$$

where the attention coefficients $\tilde{w}_1] \in \mathbb{R}^{2d'}$ and other weights are both fixed. After the two rounds of message passing, the final representations of variables $\mathbb{X}_j$ are passed through a 2-layer MLP with 64 hidden units per layer to obtain a scalar value for each variable. Finally, we apply the sigmoid function to get a calibration score between 0 and 1.

### E.2 GRAPH CONVOLUTIONAL NETWORK

#### E.2.1 STAGE I

The Network takes as input the state $s^t$ and outputs a score vector $F_\theta(s^t) \in [0, 1]^n$, one score per variable. We use 2-layer MLPs with 64 hidden units per layer and ReLU as the activation function to map each node feature and edge feature to $\mathbb{R}^d$ where $d = 64$.

Let $\mathbb{X}_j, \mathbb{B}_i, \mathbb{E}_{i,j} \in \mathbb{R}^d$ be the embeddings of the $j$-th variable, $i$-th constraint and the edge connecting them output by the embedding layers. We perform two rounds of message passing through a bipartite GCN. Denote by $\hat{A} = \hat{D}^{-\frac{1}{2}}(A+I)\hat{D}^{-\frac{1}{2}}$ the symmetrically normalized adjacency (with self-loops), and by $\hat{a}_{ij} = \hat{A}_{ij}$ its $(i, j)$-entry. In the first round (constraint update), each constraint node $\mathbb{B}_i$ aggregates its neighbors $N_i$:

$$\mathbb{B}'_i = \sigma\left(\theta_{b,1}\mathbb{B}_i + \sum_{j \in N_i} \hat{a}_{ij}(\theta_{x,1}\mathbb{X}_j + \theta_{e,1}\mathbb{E}_{i,j})\right), \tag{67}$$

where $\theta_{b,1}, \theta_{x,1}, \theta_{e,1} \in \mathbb{R}^{d \times d}$ are learnable weights and $\sigma(\cdot)$ denotes ReLU. In the second round (variable update), each variable node aggregates updated constraints:

$$\mathbb{X}'_j = \sigma\left(\theta_{x,2}\mathbb{X}_j + \sum_{i \in N_j} \hat{a}_{ji}(\theta_{b,2}\mathbb{B}'_i + \theta_{e,2}\mathbb{E}_{i,j})\right), \tag{68}$$

where $\theta_{b,2}, \theta_{x,2}, \theta_{e,2} \in \mathbb{R}^{d \times d}$ are learnable. After the two rounds of message passing, the final representations of variables $\mathbb{X}'_j$ are passed through a 2-layer MLP with 64 hidden units per layer to obtain a scalar value for each variable. Finally, we apply the sigmoid function to get a score between 0 and 1.

#### E.2.2 STAGE II

After yielding a neighborhood selection indicator $h^t \in \{0, 1\}^n$, the Network uses 2-layer MLPs with 16 hidden units per layer and ReLU to map $h^t$ to $\tilde{\mathbb{X}}_j \in \mathbb{R}^{d'}$ where $d' = 16$. We regard $\tilde{\mathbb{X}}_j$ as an extra feature and concatenate it with the original input: $[\mathbb{X}_j||\tilde{\mathbb{X}}_j]$. We perform one round of message passing reusing the *first* GCN direction (constraint update), with the original weights frozen on the original channels and new learnable adapters on the new channels:

$$\mathbb{B}'^{\text{II}}_i = \sigma\left(\theta_{b,1}\mathbb{B}_i + \sum_{j \in N_i} \hat{a}_{ij}\left(\underbrace{\theta_{x,1}\mathbb{X}_j}_{\text{frozen}} + \underbrace{\tilde{\theta}_{x,1}\tilde{\mathbb{X}}_j}_{\text{learnable}} + \theta_{e,1}\mathbb{E}_{i,j}\right)\right), \tag{69}$$

where $\theta_{b,1}, \theta_{x,1}, \theta_{e,1}$ are fixed (copied from Stage I) and $\tilde{\theta}_{x,1} \in \mathbb{R}^{d \times d'}$ is learnable. The embedding $\mathbb{B}'^{\text{II}}_i$ is then fed to a 2-layer MLP with 64 hidden units and ReLU to obtain two scalar values per constraint for predicting effective/redundant constraint features. The newly introduced learnable parameters are updated by calibrating against the constraint features during training.

### E.2.3 STAGE III

Denote the constraint feature as $r^t \in \{0,1\}^{2 \times m}$. The Network uses 2-layer MLPs with 16 hidden units per layer and ReLU to map $r^t$ to $\tilde{\mathbb{B}}_i \in \mathbb{R}^{d'}$ where $d' = 16$. We regard $\tilde{\mathbb{B}}_i$ as an extra feature and concatenate it with the original input: $[\mathbb{B}'_i \| \tilde{\mathbb{B}}_i]$. We perform one round of message passing reusing the *second* GCN direction (variable update), again freezing the original channels and learning the adapters:

$$\mathbb{X}'^{\text{III}}_j = \sigma \left( \underbrace{\theta_{x,2}\, \mathbb{X}_j}_{\text{frozen}} + \underbrace{\tilde{\theta}_{x,2}\, \tilde{\mathbb{X}}_j}_{\text{learnable}} + \sum_{i \in N_j} \hat{a}_{ji} \Big( \underbrace{\theta_{b,2}\, \mathbb{B}'_i}_{\text{frozen}} + \underbrace{\tilde{\theta}_{b,2}\, \tilde{\mathbb{B}}_i}_{\text{learnable}} + \theta_{e,2}\, \mathbb{E}_{i,j} \Big) \right), \qquad (70)$$

where $\theta_{b,2}, \theta_{x,2}, \theta_{e,2}$ are fixed (from Stage I), and $\tilde{\theta}_{x,2}, \tilde{\theta}_{b,2} \in \mathbb{R}^{d \times d'}$ are learnable. After this round, the final variable representations $\mathbb{X}'^{\text{III}}_j$ are passed through a 2-layer MLP with 64 hidden units per layer to obtain a scalar value for each variable. Finally, we apply the sigmoid function to get a calibration score between 0 and 1.

## F  DETAILS OF INSTANCE GENERATION

We present the ILP formulations for the minimum vertex cover (MVC), maximum independent set (MIS), set covering (SC) and combinatorial auction (CA) problems. MVC-S instances are generated according to the Barabasi-Albert random graph model Albert & Barabási (2002), with 1,000 nodes and average degree 70 following Song et al. (2020). MIS-S instances are generated according to the Erdos-Renyi random graph model Erdos et al. (1960), with 6,000 nodes and average degree 5 following Song et al. (2020). CA-S instances are generated with 2,000 items and 4,000 bids according to the arbitrary relations in Leyton-Brown et al. (2000). SC-S instances are generated with 4,000 variables and 5,000 constraints following Wu et al. (2021). We then generate another test set of 100 *large instances* for each problem by doubling the number of variables, namely MVC-L, MIS-L, CA-L and SC-L. For each test set, Table **??** shows its average numbers of variables and constraints. More details of instance generation are included in Appendix. For data collection and training, we generate another set of 1,024 small instances for each problem. We split these instances into training and validation sets, each consisting of 896 and 128 instances, respectively.

Table 2: Names and the average numbers of variables and constraints of the test instances.

| Name | MVC-S | MIS-S | CA-S | SC-S | MVC-L | MIS-L | CA-L | SC-L |
|---|---|---|---|---|---|---|---|---|
| #Variables | 1,000 | 6,000 | 4,000 | 4,000 | 2,000 | 12,000 | 8,000 | 8,000 |
| #Constraints | 65,100 | 23,861 | 2,422 | 5,000 | 135,100 | 48,031 | 5,221 | 5,000 |

### F.1  MINIMUM VERTEX COVER(MVC)

In an MVC instance, we are given an undirected graph $G = (V, E)$. The goal is to select the smallest subset of nodes such that at least one end point of every edge in the graph is selected:

$$\min \sum_{v \in V} x_v$$
$$\text{s.t.} \quad x_u + x_v \geq 1, \ \forall (u,v) \in E,$$
$$x_v \in \{0,1\}, \ \forall v \in V.$$

### F.2  MAXIMUM INDEPENDENT SET(MIS)

In an MIS instance, we are given an undirected graph $G = (V, E)$. The goal is to select the largest subset of nodes such that no two nodes in the subsets are connected by an edge in $G$:

$$\min - \sum_{v \in V} x_v$$
$$\text{s.t.} \quad x_u + x_v \leq 1, \ \forall (u,v) \in E,$$
$$x_v \in \{0,1\}, \ \forall v \in V.$$

## F.3  SET COVERING(SC)

In an SC instance, we are given $m$ elements and a collection $S$ of $n$ sets whose union is the set of all elements. The goal is to select a minimum number of sets from $S$ such that the union of the selected set is still the set of all elements:

$$\min \sum_{s \in S} x_s$$
$$\text{s.t.} \quad \sum_{s \in S : i \in s} x_s \geq 1, \ \forall i \in [m],$$
$$x_s \in \{0,1\}, \ \forall s \in S.$$

## F.4  COMBINATORIAL AUCTION(CA)

In a CA instance, we are given $n$ bids $\{(B_i, p_i) : i \in [n]\}$ for $m$ items, where $B_i$ is a subset of items and $p_i$ is its associated bidding price. The objective is to allocate items to bids such that the total revenue is maximized:

$$\min - \sum_{i \in [n]} p_i x_i$$
$$\text{s.t.} \quad \sum_{i : j \in B_i} x_i \leq 1, \ \forall j \in [m],$$
$$x_i \in \{0,1\}, \ \forall i \in [n].$$

## F.5  SUBSET OF MIPLIB

We construct a subset of MIPLIB (Gleixner et al., 2021) to evaluate the solvers' ability to handle challenging real-world instances. Specifically, we select instances based on their similarity, which is measured by 100 human-designed features (Gleixner et al., 2021). Instances with presolving times exceeding 300 seconds or those that exceed GPU memory limits during the inference process are discarded. Inspired by the IIS dataset used in Wang et al. (2024), we develop a refined IIS dataset containing eleven instances. We divide this dataset into training and testing sets, comprising eight training instances and three testing instances (ramos3, scpj4scip, and scpl4). Detailed information on the IIS dataset can be found in Table 3.

Table 3: Statistical information of the instances in the constructed IIS dataset.

| Instance Name | Constraint Number | Variable Number | Nonzero Coefficient Number |
|---|---|---|---|
| ex1010-pi | 1468 | 25200 | 102114 |
| fast0507 | 507 | 63009 | 409349 |
| glass-sc | 6119 | 214 | 63918 |
| iis-glass-cov | 5375 | 214 | 56133 |
| iis-hc-cov | 9727 | 297 | 142971 |
| ramos3 | 2187 | 2187 | 32805 |
| scpj4scip | 1000 | 99947 | 999893 |
| scpk4 | 2000 | 100000 | 1000000 |
| scpl4 | 2000 | 200000 | 2000000 |
| seymour | 4944 | 1372 | 33549 |
| v150d30-2hopcds | 7822 | 150 | 103991 |

# G  ADDITIONAL EXPERIMENTAL RESULTS

Table 4: Primal gap (PG) (in percent), primal integral (PI) at 45 minutes runtime cutoff, averaged over 100 test instances and their standard deviations for generated instances. "↓" means the lower the better.

| | PG (%)↓ | PI↓ | PG (%)↓ | PI↓ | PG (%)↓ | PI↓ | PG (%)↓ | PI↓ |
|---|---|---|---|---|---|---|---|---|
| | MVC-S | | MIS-S | | CA-S | | SC-S | |
| BnB | 1.47±0.26 | 68.3±19.9 | 5.69±1.37 | 200.6±87.2 | 2.64±0.77 | 158.3±39.4 | 1.38±0.88 | 89.2±43.8 |
| RANDOM | 1.42±1.05 | 55.2±39.5 | 0.35±0.14 | 23.3±8.3 | 6.2±1.57 | 300.1±47.3 | 2.81±1.46 | 160.0±39.0 |
| CL-GCN | 0.34±0.29 | 16.54±13.11 | 0.31±0.17 | 25.51±10.09 | 1.24±0.97 | 89.24±48.33 | 0.56±1.18 | 46.70±23.86 |
| CE-GCN | 0.22±0.23 | 12.92±9.66 | 0.21±0.14 | 15.57±6.97 | 0.90±0.75 | 82.36±28.71 | **0.45±0.88** | **31.34±20.75** |
| CL-GAN | 0.20±0.12 | 11.78±8.94 | 0.20±0.22 | 19.69±6.83 | 0.89±0.42 | 66.54±26.62 | 0.50±0.76 | 32.87±18.74 |
| CE-GAN | **0.15±0.14** | **6.33±6.18** | **0.15±0.09** | **11.69±4.55** | **0.50±0.48** | 46.43±25.55 | 0.58±0.34 | 54.03±12.38 |
| | MVC-L | | MIS-L | | CA-L | | SC-L | |
| BnB | 2.75±0.48 | 151.4±12.9 | 6.77±1.93 | 314.6±20.0 | 3.14±2.19 | 388.5±107.3 | 2.02±1.00 | 117.0±47.8 |
| RANDOM | 0.41±0.25 | 27.5±9.1 | 0.21±0.13 | 21.1±8.1 | 5.70±0.86 | 280.5±26.4 | 3.27±2.17 | 193.1±66.0 |
| CL-GCN | 0.26±0.09 | 23.4±7.7 | 0.29±0.28 | 29.4±15.3 | 0.22±0.11 | 270.4±46.3 | 1.48±1.08 | 86.9±59.1 |
| CE-GCN | 0.22±0.19 | 20.5±14.4 | 0.26±0.16 | 27.9±9.1 | 0.23±0.15 | 282.5±66.1 | 1.42±0.52 | 85.9±26.1 |
| CL-GAN | 0.09±0.05 | 10.9±4.5 | 0.17±0.15 | 18.4±10.0 | 0.14±0.08 | 160.0±34.1 | 0.80±0.31 | 52.9±7.8 |
| CE-GAN | **0.07±0.10** | **10.1±10.3** | **0.15±0.07** | **19.4±5.8** | **0.10±0.05** | **128.2±23.3** | **0.73±0.55** | **45.1±28.2** |

Table 5: Primal gap (PG) (in percent), primal integral (PI) at 30 minutes runtime cutoff, averaged over 100 test instances and their standard deviations for generated instances. "↓" means the lower the better.

| | PG (%)↓ | PI↓ | PG (%)↓ | PI↓ | PG (%)↓ | PI↓ | PG (%)↓ | PI↓ |
|---|---|---|---|---|---|---|---|---|
| | MVC-S | | MIS-S | | CA-S | | SC-S | |
| BnB | 2.00±0.40 | 91.4±24.3 | 7.09±1.88 | 300.8±110.6 | 3.52±1.00 | 222.7±49.5 | 1.92±1.15 | 120.5±55.0 |
| RANDOM | 1.90±1.44 | 75.3±52.7 | 0.44±0.19 | 36.6±11.0 | 8.7±2.01 | 394.2±65.4 | 4.11±2.12 | 208.7±50.5 |
| CL-GCN | 0.45±0.37 | 22.94±18.45 | 0.41±0.22 | 34.62±14.69 | 1.83±1.35 | 134.35±64.20 | 0.74±1.52 | 69.08±31.31 |
| CE-GCN | 0.32±0.30 | 18.04±13.31 | 0.27±0.19 | 23.85±9.74 | 1.20±0.91 | 99.83±41.41 | **0.59±1.04** | **39.22±26.64** |
| CL-GAN | 0.27±0.15 | 15.36±11.07 | 0.28±0.27 | 25.87±8.81 | 1.14±0.51 | 93.27±37.08 | 0.67±0.92 | 46.92±25.42 |
| CE-GAN | **0.20±0.19** | **8.84±7.71** | **0.20±0.11** | **15.10±6.53** | **0.68±0.59** | **57.90±30.62** | 0.66±0.43 | 68.15±14.25 |
| | MVC-L | | MIS-L | | CA-L | | SC-L | |
| BnB | 3.73±0.60 | 204.8±16.6 | 9.26±2.53 | 414.0±28.8 | 3.86±2.88 | 515.6±135.7 | 2.63±1.32 | 152.3±66.0 |
| RANDOM | 0.55±0.39 | 34.1±12.7 | 0.28±0.16 | 28.1±30.8 | 7.62±1.23 | 347.0±35.0 | 4.64±2.72 | 267.0±85.9 |
| CL-GCN | 0.37±0.12 | 30.5±11.7 | 0.41±0.35 | 42.0±19.3 | 0.32±0.14 | 393.5±54.1 | 1.88±1.43 | 119.3±71.0 |
| CE-GCN | 0.31±0.25 | 25.8±19.1 | 0.34±0.21 | 37.9±11.9 | 0.31±0.20 | 376.2±94.7 | 1.97±0.68 | 118.2±32.6 |
| CL-GAN | 0.11±0.07 | 14.3±6.3 | 0.21±0.21 | 25.0±12.5 | 0.18±0.12 | 226.1±47.1 | 1.06±0.39 | 70.0±10.4 |
| CE-GAN | **0.10±0.14** | **12.5±13.0** | **0.19±0.10** | **25.5±7.7** | **0.15±0.07** | **164.0±28.8** | **0.94±0.65** | **60.0±37.5** |

Table 6: Primal gap (PG) (in percent), primal integral (PI) at 45 minutes runtime cutoff. "↓" means the lower the better.

| | PG (%)↓ | PI↓ | PG (%)↓ | PI↓ |
|---|---|---|---|---|
| | WA | | IP | |
| BnB | 0.49±0.12 | 3.81±0.94 | 35.7±3.7 | 7.38±0.63 |
| RANDOM | 0.40±0.10 | 3.18±0.84 | 42.4±5.3 | 9.80±0.81 |
| CL-GCN | 0.48±0.18 | 3.40±1.24 | 30.6±3.5 | 6.87±0.53 |
| CE-GCN | 0.17±0.08 | 1.60±0.46 | 22.9±4.6 | 6.17±0.79 |
| CL-GAN | 0.29±0.21 | 2.26±1.66 | 32.2±2.5 | 8.07±0.37 |
| CE-GAN | **0.13±0.10** | **1.07±0.72** | **23.9±4.2** | **5.20±0.78** |

Table 7: Primal gap (PG) (in percent), primal integral (PI) at 30 minutes runtime cutoff. "↓" means the lower the better.

| | PG (%)↓ | PI↓ | PG (%)↓ | PI↓ |
|---|---|---|---|---|
| | WA | | IP | |
| BnB | 0.68±0.16 | 5.18±1.26 | 48.0±5.1 | 10.32±0.86 |
| RANDOM | 0.54±0.14 | 4.35±1.13 | 57.3±7.1 | 13.37±1.09 |
| CL-GCN | 0.65±0.24 | 4.66±1.69 | 41.5±4.7 | 9.59±0.73 |
| CE-GCN | 0.24±0.11 | 2.18±0.63 | 31.8±6.3 | 8.29±1.06 |
| CL-GAN | 0.39±0.29 | 3.11±2.27 | 43.4±3.4 | 10.86±0.51 |
| CE-GAN | **0.18±0.13** | **1.50±0.97** | **32.5±5.7** | **7.26±1.07** |

