# OpenReview forum: "VARIABLE COUPLING-ENHANCED LARGE NEIGHBORHOOD SEARCH FOR SOLVING INTEGER LINEAR PROGRAMS"
_ICLR.cc/2026/Conference — ICLR 2026 Conference Withdrawn Submission_

### Official Review · Reviewer_ajVq · 2025-10-28

**Soundness:** 3
**Presentation:** 1
**Contribution:** 2
**Rating:** 2
**Confidence:** 4

**Summary:**

This work challenges the independence assumption in the previous neural large neighborhood search (LNS) solvers in missing the variable coupling. It therefore proposes to augment the prior neural LNS solvers with graph decomposition, and empirically evaluation verifies the improvement on a diverse set of integer linear programs.

**Strengths:**

1. The proposed method casts the attention in the variable decoupling and redundant constraints in LNS, showing a certain level of novelty.
2. The authors have conducted a comprehensive evaluation on a wide range of ILPs, including 2 real-world benchmarks.

**Weaknesses:**

1. The experimental results are not convincing.
- CL-GAT (baseline) shows a clear advantage against the proposed method CE-GAT in the primal integral in Figure 4 (MVC-L, MIS-S and SC-L), but the results in Table 1 still indicates CL-GAT is worse than CE-GAT. This is inconsistent.
- For most cases (e.g., MVC-L, MIS-S, MIS-L, CA-L, SC-S, SC-L), CE-GAT only outperforms CL-GAT in the primal gat at the end of the 1-hour solving time budget. First, this indicates a sign of heavy parameter tuning on CE-GAT. Second, how could authors claim **a significantly better anytime performance** when CE-GAT is worse than CL-GAT in almost the first 50 minutes?
- The advantage pattern of CE-GCN/GAT against CL-GAT/GCN is very random, sometimes converging faster and sometimes converging until the end, it is very hard to make any conclusion in how the variable decoupling help the model.
- Only Random and contrastive learning baselines are considered. What about the supervised and reinforcement learning baselines in Huang, 2023? Given that you mention the other works in improving the expressive power of GNNs in Appendix A, why there is no comparison?
 2. All theoretical analyses only present trivial conclusions. Clearly, the independent assumption is not perfect and there exists the example that this assumption fails. However, considering the randomness of LNS itself, **to what extent does this limitation hurt the model performance is not answered**. From the minor improvement in Figure 4, this is not the major limitation of current neural LNS methods.
3. The presentation of this work needs significant improvement, or in other words, this work is not really complete.
- This works contains bunch of typos and format errors (like the compilation errors indicated by ?, in Theorem 5.3, Appendix A, ..). The name of GAN and GAT is also inconsistent in the Table 1 and Figure 4.
- The fontsizes in the figures are very small, no explanation in the x- and y- axis. Moreover, the captions are not informative (e.g., first ablation and second ablation). Figure 5 mixes the table and figures, not well-organized.

**Questions:**

See the weaknesses part

---

### Official Review · Reviewer_Riv8 · 2025-10-28

**Soundness:** 2
**Presentation:** 2
**Contribution:** 3
**Rating:** 4
**Confidence:** 4

**Summary:**

This paper identifies a key limitation in existing neural Large Neighborhood Search (LNS) methods for integer programming: the independence assumption in neighborhood selection ignores variable coupling, leading to suboptimal exploration. To address this, the authors propose Coupling-Enhanced Neural LNS (CE-LNS), which integrates graph decomposition with GNN-based prediction to explicitly model variable coupling and calibrate neighborhood selection accordingly. Theoretically, CE-LNS can identify redundant constraints and refine neighborhoods toward optimality; empirically, it outperforms prior neural LNS approaches across diverse ILP benchmarks, showing improved ability to escape local optima.

**Strengths:**

- The paper provides a thoughtful critique of current neural LNS frameworks and introduces CE-LNS, a coupling-aware extension that addresses a genuine gap in modeling variable interactions. The approach appears novel and tackles a meaningful limitation, with empirical results suggesting practical promise.
- The theoretical analysis is thorough, and the appendix includes extensive proofs and supplementary experiments, making the submission notably comprehensive and well-supported.

**Weaknesses:**

- Critical training details are missing, particularly regarding Phases 2 and 3 in Figure 3. It is unclear what learning paradigm is used (e.g., supervised learning, imitation learning), what constitutes the expert policy or supervision signal, and how these phases relate to Phase 1 (contrastive learning). Given that these phases form a core part of the methodological novelty, their omission makes it difficult to assess the true contribution and reproducibility of the work.
- Section 4’s theoretical development appears closely related to prior works on GNN expressiveness, specifically [1, 2], The authors should clarify the connections and distinctions between their analysis and these foundational results

[1] Xu K, Hu W, Leskovec J, et al. How Powerful are Graph Neural Networks? ICLR.
[2] Chen Z, Liu J, Wang X, et al. On Representing Linear Programs by Graph Neural Networks. ICLR.

- The experimental evaluation is limited in scope. The baselines focus primarily on variants of GCN-CL and omit recent strong neural LNS or heuristic methods, such as [3, 4], many of which outperform even Gurobi on specific instances. Moreover, comparisons against commercial solvers (e.g., Gurobi) or stronger classical heuristics are absent, weakening the empirical claims.

[3] Apollo-MILP: An Alternating Prediction-Correction Neural Solving Framework for Mixed-Integer Linear Programming, ICLR 2025.
[4] Light-milpopt: Solving large-scale mixed integer linear programs with lightweight optimizer and small-scale training dataset, ICLR 2024.

**Questions:**

- Could the authors please provide detailed descriptions of the training procedures for Phases 2 and 3 in Figure 3, including the learning objective, supervision source, and dependency (if any) on Phase 1?
- In Section 2, the definition of “redundant constraints” appears to diverge from the standard optimization definition—where a constraint is redundant if its removal does not change the feasible region.   Is the paper’s usage consistent with this, or does it refer to something else (e.g., ineffective during solving)?   Clarification would help avoid confusion.
- Is the contrastive learning component in Section 3 directly adapted from Huang et al. (2023)?   If so, the relationship should be explicitly acknowledged.
- A clear forward-pass pipeline describing how the outputs of the three phases jointly determine variable selection for destruction would greatly improve methodological transparency.
- Why was a 60-minute time limit chosen for experiments?   Given that LNS is typically designed for efficient high-quality solutions, this seems unusually long compared to prior work (e.g., 200 seconds).   Does the method only show gains under very long runtimes?
- Figure 4 suggests that under shorter time budgets (e.g., on MVC and MIS), baseline methods find higher-quality feasible solutions faster than CE-LNS.  Does this indicate that CE-LNS is less efficient in the early stages of optimization?
- There appear to be citation errors at Lines 354, 746, and 748—could these be verified and corrected?

**In fact, I am somewhat hesitant in my evaluation of this paper, and the authors’ clarification of certain details would greatly help me arrive at a more comprehensive and objective assessment. I will actively engage in the rebuttal discussion and update my score accordingly**.

---

### Official Review · Reviewer_MLKg · 2025-10-29

**Soundness:** 4
**Presentation:** 2
**Contribution:** 3
**Rating:** 8
**Confidence:** 4

**Summary:**

This paper introduces CE-LNS, a new machine learning-based Large Neighborhood Search method for solving Integer Linear Programs (ILPs). It addresses a key weakness in prior neural LNS approaches: their assumption that variables can be selected for optimization independently. Theoretically, the framework can identify important constraints and approximate optimal search neighborhoods. Empirically, it outperforms existing neural LNS methods across various benchmark problems.

**Strengths:**

1. ​The paper effectively identifies a genuine and significant limitation in existing neural LNS methods: the factorized (independent) variable selection assumption. The provided counterexample (Theorem 4.1) is clear and compelling.
2. The core idea of enhancing a standard GNN with a graph decomposition to model variable coupling is novel and well-justified. The three-stage design is logical.
3. The experimental section is comprehensive. The results consistently show that CE-LNS outperforms existing neural LNS baselines (GCN-CL, GAT-CL), which are strong and relevant competitors.
4. The paper goes beyond a purely empirical contribution by providing a theoretical analysis. Theorems 5.2 and 5.3 offer formal justification for using constraints as bags and for the framework's expressive power, linking it to the ability to identify effective constraints and approximate optimal neighborhoods.

**Weaknesses:**

1. The empirical comparison is strong but could be enhanced by including a non-learnable, coupling-aware destroy heuristic as a baseline. For example, how does CE-LNS compare to a simple destroy heuristic that selects variables based on communities detected in the variable graph? This would help isolate the benefit of the learned coupling-aware policy versus a handcrafted one.
2. The description of Stage III (Calibration of Neighborhood Selection) is the most complex part of the paper and could be clarified. Especially, the greedily selected β 1 ·k t ·n t score is somewhat ambiguous to me. A more thorough explanation is preferred.

**Questions:**

1. Could you please elaborate on how the samples (h_des_t*, h_fix_t*)are generated for training the calibration stage (Stage III)? The phrase "real-world neighborhood indicator h^t" is unclear. Is this a form of expert imitation, or is it derived from the outcomes of previous LNS iterations?
2. Theorem 5.2 provides one natural decomposition (each bag is the set of variables in a constraint). Did you explore other graph decomposition algorithms for forming the bags? What are the trade-offs?

---

### Official Review · Reviewer_ET5J · 2025-10-31

**Soundness:** 3
**Presentation:** 2
**Contribution:** 2
**Rating:** 4
**Confidence:** 3

**Summary:**

This paper proposes Coupling-Enhanced Neural Large Neighborhood Search (CE-LNS) for ILPs, addressing a core flaw of current neural LNS policies: they assume variable independence and thus give indistinguishable scores to neighborhoods with very different true optimization potential. The motivation is clear. Then the authors formalize this with a toy ILP and then augment a standard bipartite-graph GNN policy with a graph-decomposition view that treats each constraint as a “bag” of variables.

**Strengths:**

The idea of this paper is interesting and worth to explore.

**Weaknesses:**

•	Theorem 5.3 is framed as “approximate optimal neighborhoods,” but it’s conditioned on k_t < n/2 to “break symmetry,” a restrictive and somewhat ad-hoc requirement. Why you have this condition? and also the proof text leans on approximation over finite datasets / UA-style arguments rather than offering generalization guarantees. It even contains the broken “equation ??”.
•	The novelty of graph decomposition is unclear. The “bags of variables = constraints” construction is essentially the standard ILP bipartite view (constraints as factor/bag nodes). Theorem 5.2 proves this decomposition property, but given the prior bipartite representation, it reads as redundant and doesn’t yet explain why the added Stage-II messages capture new coupling beyond a deeper 1-WL-bounded GNN.
•	For the experiment, you cite RL-LNS and IL-LNS but don’t report them empirically; only Random, BnB, CL-GCN/GAT, and your CE variants are shown. Including strong recent neural solvers (e.g., RL-LNS, IL-LNS; even Apollo-MILP where applicable) would strengthen claims.
•	Figure 3 (and the surrounding text) introduces “Semi-Upper Convolution” and “Semi-Lower Convolution,” but these operations are never rigorously defined (no equations, no pseudocode)
•	Why the tables have GAN? It should be GAT..
•	Stage III calibration rule is unclear. The text says CE-LNS “selects β₁·k_t·n_t variables,” where n_t is “the proportion” of overlap between two neighborhoods; mixing a count (k_t) with a proportion (n_t) is unclear and can yield a non-integer selection size.
•	Multiple misspellings and style glitches (“Variale/Constrain,” “GAN” for GAT in tables), and several “Table ??/Figure ??/equation ??” placeholders remain. These erode trust in results.

**Questions:**

How is n_t defined (count vs. proportion), how are non-integers rounded, and how are ties resolved? Is the “destroy/fix” split symmetric by design or tuned?

For Theorem 5.3, please justify the choice of  k_t < n/2 .

What losses train Stage-I/II/III? If you predict “effective vs. redundant” constraints, how are labels created? Any class imbalance handling?

I do not see a strong link between the theory and the practical part. A mapping from theory assumption to experiment implementation would be helpful.

---

### Note · Authors · 2025-11-26

**Comment:**

Dear Area Chair and Reviewers,

We are writing to officially withdraw our submission 12765, titled "Coupling-Enhanced Neural Large Neighborhood Search (CE-LNS) for ILPs".We would like to sincerely thank all the reviewers for their time and extensive feedback. Although the decision to withdraw is difficult, we believe it is the right step to ensure the quality of this work meets the highest standards.

**Withdrawal Confirmation:**

I have read and agree with the venue's withdrawal policy on behalf of myself and my co-authors.